# Global Identifiability of $\ell_1$-based Dictionary Learning via Matrix Volume Optimization

**Jingzhou Hu**    **Kejun Huang**

Department of Computer and Information Science and Engineering
University of Florida
Gainesville, FL 32611
(jingzhouhu,kejun.huang)@ufl.edu

## Abstract

We propose a novel formulation for dictionary learning that minimizes the determinant of the dictionary matrix, also known as its volume, subject to the constraint that each row of the sparse coefficient matrix has unit $\ell_1$ norm. The main motivation for the proposed formulation is that it provides global identifiability guarantee of the groundtruth dictionary and sparse coefficient matrices, up to the inherent and inconsequential permutation and scaling ambiguity, if a set of vectors obtained from the coefficient matrix lies inside the $\ell_\infty$ norm ball but contains the $\ell_2$ norm ball in their convex hull. Unlike existing work on identifiability of dictionary learning, our result is global, meaning that a globally optimal solution to our proposed formulation has to be a permuted and rescaled version of the groundtruth factors. Another major improvement in our result is that there is no additional assumption on the dictionary matrix other than it is nonsingular, unlike most other works that require the atoms of the dictionary to be mutually incoherent. We also provide a probabilistic analysis and show that if the sparse coefficient matrix is generated from the widely adopted Bernoulli-Gaussian model, then it is globally identifiable if the sample size is bigger than a constant times $k \log k$, where $k$ is the number of atoms in the dictionary, with overwhelming probability. The bound is essentially the same as those local identifiability results, but we show that it is also global. Finally, we propose algorithms to solve the new proposed formulation, specifically one based on the linearized-ADMM with efficient per-iteration updates. The proposed algorithms exhibit surprisingly effective performance in correctly and efficiently recovering the dictionary, as demonstrated in the numerical experiments.

## 1   Introduction

Dictionary learning is a famous unsupervised learning problem [Tošić and Frossard, 2011] that also goes by the name sparse coding [Olshausen and Field, 1997] or sparse component analysis [Georgiev et al., 2005]. It stems from the intuition that samples from a data set can all be approximately represented as a sparse combination of atoms from a complete or overcomplete dictionary, or $\boldsymbol{x}_i = \boldsymbol{A}\boldsymbol{s}_i$ for $i = 1, \ldots, n$, where $\boldsymbol{x}_i \in \mathbb{R}^m$ are the given data points, columns of $\boldsymbol{A} \in \mathbb{R}^{m \times k}$ represents atoms of a dictionary, and $\boldsymbol{s}_i \in \mathbb{R}^k$ are the sparse representations of $\boldsymbol{x}_i$, indicating that $\boldsymbol{x}_i$ is linearly dependent on only a few atoms from the dictionary $\boldsymbol{A}$ with coefficients given in the nonzeros of $\boldsymbol{s}_i$. Staking $\boldsymbol{x}_i$ as columns of $\boldsymbol{X} \in \mathbb{R}^{m \times n}$ and $\boldsymbol{s}_i$ as columns of $\boldsymbol{S} \in \mathbb{R}^{k \times n}$, it can be written as a matrix factorization model $\boldsymbol{X} = \boldsymbol{A}\boldsymbol{S}$. Although in some cases there exist known dictionaries that admit good-quality sparse representations, such as the DCT or wavelet basis for image compression, sometimes it is better to learn a sparse-inducing dictionary from data in an unsupervised fashion, hence the task of dictionary learning that tries to estimate $\boldsymbol{A}$ and $\boldsymbol{S}$ jointly. In most cases, people

37th Conference on Neural Information Processing Systems (NeurIPS 2023).

assume $m \leq k$: the dictionary is complete if $m = k$, and overcomplete if $m < k$. In this paper we focus on the complete dictionary case when $\boldsymbol{A}$ is square and nonsingular, which is common in a lot of prior works that this work is built upon [Gribonval and Schnass, 2010, Sun et al., 2016a,b, Wu and Yu, 2017, Wang et al., 2020]

Dictionary learning has found numerous applications in signal denoising [Elad and Aharon, 2006], audio coding [Plumbley et al., 2009], and medical imaging [Tošić et al., 2010], to name just a few. On the theory side, most of the existing works have focused on algorithm design. Famous algorithms include $k$-SVD [Aharon et al., 2006a] and online dictionary learning [Mairal et al., 2009], among numerous other algorithms based on generic nonconvex algorithm design with guarantee of convergence to a stationary point. More recently, there have appeared a line of research that attempts to show global optimality for dictionary learning under more restrictive assumptions, such as [Spielman et al., 2012, Agarwal et al., 2016, Arora et al., 2014, 2015, Sun et al., 2016a,b, Rambhatla et al., 2019, Bai et al., 2019, Zhai et al., 2020a,b, Shen et al., 2020, Tolooshams and Ba, 2022].

## 1.1 Model identifiability

The dictionary learning problem can be considered an instance of the broader matrix factorization model $\boldsymbol{X} = \boldsymbol{A}\boldsymbol{S}$ subject to the opaque requirement that $\boldsymbol{S}$ is sparse. It is well-known that matrix factorization without any additional assumptions on the latent factors is not unique, since we can always "insert" an invertible matrix $\boldsymbol{Q}$ and $\boldsymbol{Q}^{-1}$ as $\boldsymbol{X} = \widetilde{\boldsymbol{A}}\widetilde{\boldsymbol{S}}$ where $\widetilde{\boldsymbol{A}} = \boldsymbol{A}\boldsymbol{Q}$ and $\widetilde{\boldsymbol{S}} = \boldsymbol{Q}^{-1}\boldsymbol{S}$, and one cannot distinguish whether $\boldsymbol{S}$ or $\widetilde{\boldsymbol{S}}$ are the groundtruth sources. Such rotation ambiguity cannot be resolved by the well-known principal component analysis (PCA) [Jolliffe, 2002].

On the other hand, most matrix factorization models allow scaling and permutation ambiguity, i.e., a permutation matrix $\boldsymbol{\Pi}$ and a diagonal matrix $\boldsymbol{D}$ such that the recovered mixing matrix is $\boldsymbol{A}\boldsymbol{D}\boldsymbol{\Pi}$ and the recovered sources are $\boldsymbol{\Pi}^{\top}\boldsymbol{D}^{-1}\boldsymbol{S}$. In addition to inherent constraints that fit the application, it is sometimes helpful to have an optimization criterion in search for the "correct" factorization (up to scaling and permutation). In light of this, we formally define the identifiability of any matrix factorization model as follows:

**Definition 1.1** (Identifiability). Consider the generative model $\boldsymbol{X} = \boldsymbol{A}^{\natural}\boldsymbol{S}^{\natural}$, where $\boldsymbol{A}^{\natural}$ and $\boldsymbol{S}^{\natural}$ are the groundtruth latent factors. Let $(\boldsymbol{A}^{\star}, \boldsymbol{S}^{\star})$ be optimal for an identification criterion $q$

$$(\boldsymbol{A}^{\star}, \boldsymbol{S}^{\star}) = \underset{\substack{\boldsymbol{X}=\boldsymbol{A}\boldsymbol{S} \\ \boldsymbol{A}\in\mathcal{A}, \boldsymbol{S}\in\mathcal{S}}}{\arg\min}\ q(\boldsymbol{A}, \boldsymbol{S}),$$

where $\mathcal{A}$ and $\mathcal{S}$ are some constraint set that $\boldsymbol{A}$ and $\boldsymbol{S}$ should belong to, respectively. If $\boldsymbol{A}^{\natural}$ and/or $\boldsymbol{S}^{\natural}$ satisfy some condition such that for any $(\boldsymbol{A}^{\star}, \boldsymbol{S}^{\star})$, there exist a permutation matrix $\boldsymbol{\Pi}$ and a diagonal matrix $\boldsymbol{D}$ such that $\boldsymbol{A}^{\natural} = \boldsymbol{A}^{\star}\boldsymbol{D}\boldsymbol{\Pi}$ and $\boldsymbol{S}^{\natural} = \boldsymbol{\Pi}^{\top}\boldsymbol{D}^{-1}\boldsymbol{S}^{\star}$, then we say that the matrix factorization model is essentially identifiable, up to permutation and scaling, under that condition.

For dictionary learning in specific, where the essential constraint on the latent factors is sparsity on $\boldsymbol{S}$, there have been some existing works on its identifiability issue. It has been shown that simply by putting a hard sparsity constraint on $\boldsymbol{S}$ is enough to guarantee a unique factorization, provided that the sample size $n$ is large enough [Aharon et al., 2006b, Hillar and Sommer, 2015, Cohen and Gillis, 2019]. The main drawback is that the required sample size $n$ is usually too large to be practical—results from [Aharon et al., 2006b] and [Hillar and Sommer, 2015] both require $n$ to be $O((k+1)\binom{k}{s})$, where $s$ is the number of nonzeros that each column of $\boldsymbol{S}$ is allowed to have; it has been relaxed to $O(k^3/(k-s)^2)$ by Cohen and Gillis [2019], but still very large. The existence of $s$ in the bound also shows that an outlier sample that does not admit a sparse representation from the dictionary is absolutely not allowed in the data set, which is also restrictive. Inspired by the success of compressive sensing and sparse recovery, it is perhaps more common to use the $\ell_1$ norm to approximately sparsify $\boldsymbol{S}$. Identifiability results based on the $\ell_1$ norm formulation has been predominantly local, meaning the model is identifiable within a neighborhood of the groundtruth factors $(\boldsymbol{A}^{\natural}, \boldsymbol{S}^{\natural})$, but on the bright side the sample size requirement is typically down to $O(k \log k)$ and allows the existence of dense outliers [Gribonval and Schnass, 2010, Wu and Yu, 2017, Wang et al., 2020].

## 1.2 This paper

In this paper, we intend to bridge the gap by providing *global* identifiability guarantee of complete dictionary recovery using a $\ell_1$ norm based formulation, given as follows:

$$\underset{A,S}{\text{minimize}} \quad |\det A| \qquad \text{subject to} \quad X = AS, \|S_{j,:}\|_1 \leq 1, j = 1, \ldots, k, \tag{1}$$

To the best of our knowledge, this is a novel formulation that has never appeared in the literature; however, as we will show in the next section, the intuition stems from the more commonly used $\ell_1$-based formulations. We stress that the formulation is $\ell_1$-based, as the $\ell_1$ norm is the most widely used term to promote sparsity of the solution. Although in the context of dictionary learning, using the $\ell_1$ norm as the convex surrogate for sparsity is not enough to make the overall problem tractable, there are still benefits of using it instead of other ones such as hard sparsity constraints, for example in terms of sample complexity as introduced in the previous subsection.

After some gentle derivation to motivate the proposed new formulation (1) for dictionary learning, we provide the following contributions:

1. We give a deterministic characterization of the identifiability condition on the sparse coefficient matrix $S$ under which the dictionary (as well as the sparse coefficients) can be uniquely determined, up to the inherent and inconsequential permutation and scaling ambiguity, via solving (1). Unlike existing work that shows local identifiability, our result is global, meaning that the groundtruth factors have to be a global minimizer to (1). Furthermore, perhaps surprisingly, our result only require the complete dictionary to be nonsingular—there are no additional assumptions that require the atoms to be mutually incoherent whatsoever.

2. The result shows that $\ell_1$-based dictionary learning is globally identifiable if a certain set of points, which are by definition inside the hypercube $[-1, 1]^k$, contains the unit ball in their convex hull. This geometric assumption is called "sufficiently scattered", with similar ideas appearing in pertinent unsupervised learning models. We provide intuition why this assumption generally leads to a sparse $S$. We further show that if the sparse coefficient follows the widely used Bernoulli-Gaussian generative model, then it is identifiable with very high probability, provided the sample size $n$ is bigger than a constant times $k \log(k)$. This is the same sample complexity bound obtained in some of the local identifiability analysis, but we show that it is also global, using the proposed new formulation (1).

3. We propose algorithms to solve (1), specifically one based on linearized-ADMM in the main paper; in the supplementary we also present one based on the Frank-Wolfe framework and the other based on block coordinate descent. Even though the proposed formulation (1) is nonconvex (just like any other dictionary learning formulation), numerical experiments on synthetic data show that they achieve global optimality in almost all cases, provided that the groundtruth latent factors satisfy the identifiability assumptions given before.

## 2 Problem Formulation

We start by proposing a novel formulation for dictionary learning. The purpose is to show that the seemingly different formulation does not come out of thin air, but is highly related, if not equivalent, to existing ones that have been studied extensively. Consider the following formulation adopted in [Gribonval and Schnass, 2010, Geng and Wright, 2014, Wu and Yu, 2017, Wang et al., 2020]:

$$\underset{A,S}{\text{minimize}} \quad \|S\|_1 \qquad \text{subject to} \quad X = AS, \|A_{:,j}\| \leq 1, j = 1, \ldots, k. \tag{2}$$

We use the noiseless model $X = AS$ since the main focus of this work is identifiability, and a lot of algorithms are designed by replacing this term with a loss function that measures the residual $X - AS$, e.g. [Gribonval et al., 2015]. The purpose of the $\ell_1$ norm of $S$ is to encourage it to be sparse, but since it is also dependent on the scaling of $S$, the constraint $\|A_{:,j}\| \leq 1$ is imposed onto the columns of $A$. Such constraints are without loss of generality due to the scaling ambiguity $AS = ADD^{-1}S$, so for any admissible factorization, we can rescale the columns of $A$ to unit norm, while absorbing the scaling into the rows of $S$, without affecting the matrix product.

Due to the Lagrangian multiplier theorem, there exist positive scalars $\beta_1, \ldots, \beta_k$ and $\gamma_1, \ldots, \gamma_k$ such that any solution of the following problem is also optimal for (2)

$$\underset{A,S}{\text{minimize}} \quad \sum_{j=1}^{k} \gamma_j \|A_{:,j}\| \qquad \text{subject to} \quad X = AS, \|S_{j,:}\|_1 \le \beta_j, j = 1, \dots, k, \qquad (3)$$

since they would lead to exactly the same Lagrangian function with appropriate choices of $\gamma_j$ and $\beta_j$. Again, due to the inherent scaling ambiguity, we can assume without loss of generality that $\beta_1 = \cdots = \beta_k = 1$, leaving the unknown parameters to just $\gamma_j$'s.

If we replace the summation in the objective of (3) with product, or equivalently replacing the arithmetic mean with the geometric mean, the term $\prod_j \gamma_j \|A_{:,j}\|$ would be intimately related to $|\det A|$ if $A$ is square. Indeed, there must exist some unknown matrix $G$ such that $|\det GA| = \prod_j \gamma_j \|A_{:,j}\|$. Furthermore, for the purpose of optimization it is completely unnecessary to know $G$ since $|\det GA| = |\det G||\det A|$, which is just a positive scaling of $|\det A|$ that does not affect the optimization solution. As a result, our proposed novel formulation of dictionary learning takes the form of (1).

Although our paper focuses on the case of complete dictionaries, i.e., when $A$ is square and non-singular, they can be effortlessly extended to the overdetermined case when $A$ is tall, and the only modification is to change the objective to $\det A^\top A$. This is sometimes called the volume of the matrix [Ben-Israel, 1992] with applications in change-of-variable integration [Ben-Israel, 1999] and probability [Ben-Israel, 2000]. It has been used as the identification criterion for several variants of nonnegative matrix factorization with an interesting geometric interpretation of finding the minimum volume enclosing simplex of a set of points [Lin et al., 2015, Huang et al., 2016, 2018, Fu et al., 2018, Huang and Fu, 2019]. The most related work is the polytopic matrix factorization framework proposed by Tatli and Erdogan [2021], which includes a case that constraints the *columns* of $S$ to have bounded $\ell_1$ norm. Our proposed formulation constrains the rows of $S$, which seems to be a minor difference but is in fact a great leap forward since row scaling of $S$ is inherent for any matrix factorization, while column scaling is not. It also turns out that the identifiability analysis would be drastically different, albeit the formulations look similar.

## 3 Global identifiability

In this section we study the conditions under which the proposed dictionary learning formulation (1) is identifiable, i.e., any solution to (1) equals to the groundtruth dictionary $A^\natural$ and sparse coefficients $S^\natural$ up to permutation and scaling, as per Definition 1.1. The groundtruth $S^\natural$ can have arbitrary scaling, and it is without loss of generality to assume that its rows have unit $\ell_1$ norms; we denote such rescaled $S^\natural$ to be $\widetilde{S}^\natural$, i.e., $\|\widetilde{S}^\natural_{j,:}\|_1 \le 1, j = 1, \dots, k$; we absorb the counterscaling into the columns of $A^\natural$ to generate $\widetilde{A}^\natural$, so that $X = A^\natural S^\natural = \widetilde{A}^\natural \widetilde{S}^\natural$.

Since we consider only complete dictionaries, for any feasible $(A, S)$ for (1), there must exist a nonsingular matrix $W$ such that $S = W\widetilde{S}^\natural$ and $A = \widetilde{A}^\natural W^{-1}$, therefore Problem (1) is, conceptually, equivalent to

$$\underset{W}{\text{maximize}} \quad |\det W| \qquad \text{subject to} \quad \|w_j^\top \widetilde{S}^\natural\|_1 \le 1, j = 1, \dots, k, \qquad (4)$$

where $w_j^\top$ denotes the $j$th row of $W$. Since $W = A^{-1}\widetilde{A}^\natural$ and $\widetilde{A}^\natural$ is feasible, the optimal value of (4) is at least 1. On the other hand, if the model is identifiable, then the optimal value of (4) should be exactly equal to 1, and the optimal solution to (4) must be a permutation matrix times a diagonal $\pm 1$ matrix. The rest of this section is dedicated to characterizing when such condition holds for the sparse coefficients $S$. Notice that all we need for the dictionary $A$ is that its columns are independent, so that the transform from (1) to (4) is possible—there is no additional assumptions such as "incoherence" for the rest of the arguments to hold, which is a great improvement to the existing works such as [Gribonval and Schnass, 2010, Sun et al., 2016a,b, Wu and Yu, 2017, Wang et al., 2020].

### 3.1 Deterministic characterization

The intuition is simple: the Hadamard inequality states that $|\det W| \le \prod_j \|w_j\|$, so a sufficient condition for identifiability is that the set of solutions to the following (nonconvex) quadratic programming

$$\underset{w}{\text{maximize}} \quad \|w\|^2 \quad \text{subject to} \quad \|w^\top \widetilde{S}^\natural\|_1 \le 1 \qquad (5)$$

is $\{\pm \boldsymbol{e}_1, \ldots, \pm \boldsymbol{e}_k\}$; in other words, a maximizer must be an indicator vector or its negative.

To proceed, we introduce the following notion:

**Assumption 3.1** (Sufficiently scattered in the hypercube). Let $\mathcal{B}$ denote the Euclidean ball $\mathcal{B} = \{\boldsymbol{x} \in \mathbb{R}^k \mid \|\boldsymbol{x}\| \leq 1\}$ and $C$ denote the hypercube $C = \{\boldsymbol{x} \in \mathbb{R}^k \mid \|\boldsymbol{x}\|_\infty \leq 1\}$. A set $\mathcal{S}$ is sufficiently scattered in the hypercube if:

1. $\mathcal{B} \subseteq \mathcal{S} \subseteq C$;
2. $\partial \mathcal{B} \cap \partial \mathcal{S} = \{\pm \boldsymbol{e}_1, \ldots, \pm \boldsymbol{e}_k\}$, where $\partial$ denotes the boundary of the set.

A geometric illustration of a set that is sufficiently scattered in the hypercube is shown in Figure 1, which in this case is a polytope with vertices depicted as dots. As we can see, $\mathcal{B}$ is a subset of the hypercube $[-1, 1]^k$, but touches the boundary of $[-1, 1]^k$ at points $\pm \boldsymbol{e}_1, \ldots, \pm \boldsymbol{e}_k$. If a set $\mathcal{S}$ is sufficiently scattered in $[-1, 1]^k$, $\mathcal{S}$ contains $\mathcal{B}$ as a subset and, as a second requirement, $\mathcal{B}$ touches the boundary of $\mathcal{S}$ only at those points too.

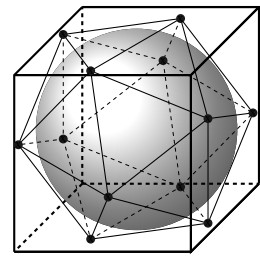

The term "sufficiently scattered" was first coined by Huang et al. [2016] to characterize the identifiability condition for nonnegative matrix factorization that has already appeared in [Huang et al., 2013]. The difference is that in [Huang et al., 2013, 2016, Fu et al., 2018], the condition is defined over the conic hull of a set of points in the nonnegative orthant containing a specific hyperbolic cone. It has also been defined over the convex hull of a set of points in the probability

Figure 1: An example of sufficiently scattered in 3D.

simplex [Fu et al., 2015, Lin et al., 2015]. The most related case is the sufficiently scattered condition defined for the hypercube ($\ell_\infty$-norm ball) [Tatli and Erdogan, 2021], as well as the standard orthoplex ($\ell_1$-norm ball).

To see how the sufficiently scattered condition could be useful for dictionary learning, we first rewrite the single constraint in (5) as, equivalently, a set of $2^n$ linear inequalities $\boldsymbol{w}^\top \widetilde{\boldsymbol{S}}^\natural \boldsymbol{v} \leq 1, \forall \boldsymbol{v} \in \{\pm 1\}^n$. Since the rows of $\widetilde{\boldsymbol{S}}^\natural$ have been rescaled to have unit $\ell_1$ norm, we must have that $\|\widetilde{\boldsymbol{S}}^\natural \boldsymbol{v}\|_\infty \leq 1$, meaning that all vectors $\widetilde{\boldsymbol{S}}^\natural \boldsymbol{v}$ should lie in the hypercube $[-1, 1]^k$, and we further assume the convex hull of all such $\widetilde{\boldsymbol{S}}^\natural \boldsymbol{v}$ is sufficiently scattered. Define the $n \times 2^n$ matrix $\boldsymbol{V}$ that contains all elements of $\{\pm 1\}^n$ as its columns, we have the following deterministic characterization of the identifiability of (1):

**Theorem 3.2.** *Consider the generative model $\boldsymbol{X} = \boldsymbol{A}^\natural \boldsymbol{S}^\natural$, where $\boldsymbol{A}^\natural \in \mathbb{R}^{k \times k}$ is the groundtruth dictionary and $\boldsymbol{S}$ is the groundtruth sparse coefficients. If $\mathrm{rank}(\boldsymbol{A}^\natural) = k$ and $\mathrm{conv}(\boldsymbol{S}^\natural \boldsymbol{V})$ is sufficiently scattered in the hypercube as in Assumption 3.1, then for any solution of (1), denoted as $(\boldsymbol{A}^\star, \boldsymbol{S}^\star)$, there exist a permutation matrix $\boldsymbol{\Pi}$ and a diagonal matrix $\boldsymbol{D}$ such that $\boldsymbol{A}^\natural = \boldsymbol{A}^\star \boldsymbol{D} \boldsymbol{\Pi}$ and $\boldsymbol{S}^\natural = \boldsymbol{\Pi}^\top \boldsymbol{D}^{-1} \boldsymbol{S}^\star$. In other words, dictionary learning is identifiable if the groundtruth $\boldsymbol{A}^\natural$ has full column rank and $\boldsymbol{S}^\natural \boldsymbol{V}$ is sufficiently scattered.*

The full proof is relegated to Appendix A. Here we provide a concise sketch of the proof: if $\widetilde{\boldsymbol{S}}^\natural \boldsymbol{V}$ is sufficiently scattered, a solution to (5) must be $\pm \boldsymbol{e}_j$; as a result, $|\det \boldsymbol{W}| \leq \prod_j \|\boldsymbol{w}_j\| \leq 1$, and equality holds only if $\boldsymbol{W}$ is a permutation matrix with some sign flips of its rows.

How is this assumption relevant to the sparsity of $\boldsymbol{S}$? Notice that if $\boldsymbol{S} \boldsymbol{V}$ is sufficiently scattered, then for *any* vector $\boldsymbol{u}$ such that $\|\boldsymbol{u}\| \leq 1$, there must exist a convex coefficient $\boldsymbol{\theta}$, i.e., $\boldsymbol{\theta} \geq 0$ and $\boldsymbol{I}^\top \boldsymbol{\theta} = 1$, such that $\boldsymbol{u} = \boldsymbol{S} \boldsymbol{V} \boldsymbol{\theta}$. Due to the definition of $\boldsymbol{V}$, we know that the vector $\boldsymbol{y} = \boldsymbol{V} \boldsymbol{\theta}$ must satisfy $\|\boldsymbol{y}\|_\infty \leq 1$. Let us instantiate it at $\boldsymbol{u} = \boldsymbol{e}_j$. This means the $j$th row of $\boldsymbol{S}$ inner product with $\boldsymbol{y}$ equals to 1. But the $j$th row of $\boldsymbol{S}$ has unit $\ell_1$ norm and $\|\boldsymbol{y}\|_\infty \leq 1$, so their inner product is 1 only if $\boldsymbol{y}^\top = \mathrm{sign}(\boldsymbol{S}_{j,:})$, which is uniquely defined if $\boldsymbol{S}_{j,:}$ is completely dense. Now $\boldsymbol{e}_j = \boldsymbol{S} \boldsymbol{y}$ also implies the inner products between $\boldsymbol{y}$ and the rest of the rows of $\boldsymbol{S}$ all equal to zero, which is almost impossible if $\boldsymbol{y}^\top = \mathrm{sign}(\boldsymbol{S}_{j,:})$ is the only candidate. On the other hand, if some of the entries in $\boldsymbol{S}_{j,:}$ are zeros, then the corresponding entries in $\boldsymbol{y}$ are free to choose any value in $[-1, 1]$; to make the rest of the $k - 1$ inner products equal to zero, we generically need at least $k - 1$ free variables in $\boldsymbol{y}$, meaning $\boldsymbol{S}_{j,:}$ contains at least $k - 1$ zeros—it is still not guaranteed that $\|\boldsymbol{y}\|_\infty \leq 1$, but it intuitively makes sense that the more zeros each row of $\boldsymbol{S}$ contains, the more likely that $\boldsymbol{e}_j = \boldsymbol{S} \boldsymbol{y}$ is feasible, which is necessary for $\boldsymbol{S} \boldsymbol{V}$ to be sufficiently scattered.

## 3.2 Probabilistic characterization

The analysis given in §3.1 gives an exact characterization of when dictionary learning is identifiable using the proposed formulation (1) with a matrix volume identification criterion. However, given some sparse coefficient matrix $S$, checking whether it satisfies the identifiability condition in Theorem 3.2 amounts to solving a nonconvex quadratic programming (5), which is NP-hard in general. In this section, we assume that the sparse coefficient matrix $S$ is generated from a probabilistic generative model, specifically the Bernoulli-Gaussian model that has been widely considered in the literature, such as [Gribonval and Schnass, 2010, Sun et al., 2016a], and show that it would satisfy guarantee identifiability with high probability, provided that the number of data points $n$ is $O(k \log k)$.

**Assumption 3.3** (Bernoulli-Gaussian model). The matrix $S \in \mathbb{R}^{k \times n}$ is generated from a Bernoulli-Gaussian model with parameter $p \in (0, 1)$, denoted as $S \sim \mathcal{BG}(p)$, if its elements are i.i.d. with $S_{ij} = b_{ij} g_{ij}$, where $b_{ij} \in \{0, 1\}$ are i.i.d. Bernoulli random variables with $\Pr[b_{ij} = 1] = p$ (thus $\Pr[b_{ij} = 0] = 1 - p$) and $g_{ij} \sim \mathcal{N}(0, 1)$ are i.i.d. standard normal random variables.

Note that due to the scaling ambiguity of matrix factorization the assumption that the normal random variable has unit variance is without loss of generality—its rows will be rescaled to have unit $\ell_1$ norms nonetheless. What we are essentially assuming is that 1) a significant portion of $S$ are zero; 2) nonzeros of $S$ have zero mean and approximately the same variance.

**Theorem 3.4.** *Suppose $S \in \mathbb{R}^{k \times n}$ is generated from the Bernoulli-Gaussian model $\mathcal{BG}(p)$, and $\widetilde{S}$ is obtained by scaling its rows to have unit $\ell_1$ norm. Then*

$$\Pr \left[ \sup_{\|\boldsymbol{w}^\top \tilde{\boldsymbol{S}}\|_1 \leq 1} \|\boldsymbol{w}\| > 1 \right] \leq 4 \exp\left(k \log(k) - np(1-p)\right). \tag{6}$$

**Corollary 3.5.** *Consider the generative model $\boldsymbol{X} = \boldsymbol{A}^\natural \boldsymbol{S}^\natural$, where $\boldsymbol{A}^\natural \in \mathbb{R}^{k \times k}$ is the groundtruth dictionary and $S$ is the groundtruth sparse coefficients. If $\mathrm{rank}(\boldsymbol{A}^\natural) = k$ and the matrix $\boldsymbol{S}^\natural \in \mathbb{R}^{k \times n}$ is generated from the Bernoulli-Gaussian model $\mathcal{BG}(p)$, then $(\boldsymbol{A}^\natural, \boldsymbol{S}^\natural)$ are globally identifiable via optimizing (1) with probability at least $1 - 4 \exp\left(k \log(k) - np(1-p)\right)$.*

The proof of Theorem 3.4 is relegated to Appendix B. It is appealing to see that the presented result looks much cleaner than prior work such as [Gribonval and Schnass, 2010, Wu and Yu, 2017, Wang et al., 2020], on top of the fact that such identifiability result is *global*, and does not require *any* incoherence assumption on the dictionary, as long as it is nonsingular. An interesting observation is that most of the foundational results have appeared in the seminal work [Gribonval and Schnass, 2010]; the difference is that they are used in the improved formulation (1), and in return leads to more significant result in Corollary 3.5. From the above result, we observe the following:

1. The right-hand-side of (6) would be a valid probability ($< 1$) if $n$ is bigger than a constant times $k \log(k) / p(1 - p)$, and very quickly approaches zero as it grows bigger. This agrees with the prior work on local identifiability that the sample complexity is of the same order [Gribonval and Schnass, 2010, Geng and Wright, 2014, Wu and Yu, 2017, Wang et al., 2020].
2. The Bernoulli parameter $p$ cannot be exactly equal to 1 or 0, which affirms the obvious fact that $S$ can neither be completely dense nor entirely zero. The more interesting implication is that DL is identifiable (with high probability) as long as there is *some* nonzero probability of having zeros in $S$, no matter how small that probability is, provided the sample size is large enough.

## 4 Proposed algorithm

In this section, we propose two algorithms for the volume-optimization based formulation (1). We will reformulate (1) as a determinant maximization problem subject to linear constraints, and then apply the Frank-Wolfe algorithm or block coordinate descent, which are both guaranteed to converge to a stationary point.

Since we assume $A$ is nonsingular, we can define $\boldsymbol{P} = \boldsymbol{A}^{-1}$ and apply a change of variable to problem (1): the objective would become minimizing $1/|\det \boldsymbol{P}|$, which we apply the log function and make it $-\log \det |\boldsymbol{P}|$, and in the constraints we can now eliminate the $S$ variables by simply requiring $\|\boldsymbol{P}\boldsymbol{X}_{j,:}\|_1 \leq 1, j = 1, \ldots, k$. This leads to the following reformulation

$$\underset{\boldsymbol{P}}{\text{minimize}} \quad -\log |\det \boldsymbol{P}| \qquad \text{subject to} \quad \|[\boldsymbol{P}\boldsymbol{X}]_{j,:}\|_1 \leq 1, j = 1, \ldots, k. \tag{7}$$

Problem (7) now has a convex, or more specifically linear, constraint set, although the objective is still not convex. In Appendix 4, we propose two algorithms to directly tackle this formulation, one based on the Frank-Wolfe framework, and the other is an instance of block coordinate descent. Both algorithms guarantee convergence to a stationary point, which is the best one can get for a generic nonconvex problem. However, both algorithms require solving a linear programming problem in each iteration, making the overall algorithm double-looped, and the time efficiency is greatly affected by the specific subroutine to solve linear programming. In the rest of this section, we propose an algorithm based on the linearized alternating direction method of multipliers (L-ADMM) with well-defined and low-complexity iterations.

## 4.1 Linearized alternating direction method of multipliers (L-ADMM)

We first modify the formulation (7) by introducing an auxiliary variable $S$:

$$\underset{P,S}{\text{minimize}} \quad -\log|\det P| \qquad \text{subject to} \quad PX = S, \ \|S_{j,:}\|_1 \leq 1, j = 1, \ldots, k. \tag{8}$$

Formulation (8) consists of two sets of variables ($P$ and $S$) over two separable functions (one of them being an indicator function of $S$ that the $\ell_1$ norms of its rows are no bigger than 1) and linear equality constraints. It is easy to derive the alternating direction method of multipliers (ADMM) [Boyd et al., 2011] for (8):

$$
\begin{cases}
P_{(t+1)} \leftarrow \arg\min_{P} \ -\log|\det P| + (\rho/2)\|PX - S_{(t)} + U_{(t)}\|^2, & \text{(9a)} \\[2mm]
S_{(t+1)} \leftarrow \arg\min_{S} \ \mathbb{1}_{\|\cdot\|_1 \leq 1}(S) + (\rho/2)\|P_{(t+1)}X - S + U_{(t)}\|^2, & \text{(9b)} \\[2mm]
U_{(t+1)} \leftarrow U_{(t)} + P_{(t+1)}X - S_{(t+1)}. & \text{(9c)}
\end{cases}
$$

The second step (9b) is well-defined, as it projects each row of the matrix $P_{(t+1)}X + U_{(t)}$ to the $\ell_1$ norm ball, which can be efficiently computed with linear complexity [Duchi et al., 2008]. In our implementation we use the bisection method described in [Parikh and Boyd, 2014]. The first step (9a), however, is not clear how to compute. One popular method to mitigate this issue, which is prevalent in ADMM, is to take a linear approximation of the loss function of $P$ at the previous iterate $P_{(t)}$ [Lu et al., 2021]. The gradient of $\log|\det P|$ is $P^{-\top}$, therefore the linear approximation of $-\log|\det P|$ at $P_{(t)}$ is $-\log|\det P_{(t)}| - \text{Tr}\, P_{(t)}^{-1}(P - P_{(t)})$. The overall update of $P_{(t+1)}$ becomes minimizing a convex quadratic function, which can be done in closed-form. The derived linearized ADMM (L-ADMM) iterates are

$$
\begin{cases}
P_{(t+1)} \leftarrow ((S_{(t)} - U_{(t)})X^\top + (1/\rho)P_{(t)}^{-\top})(XX^\top)^{-1}, & \text{(10a)} \\[2mm]
S_{(t+1)} \leftarrow \text{Proj}_{\|\cdot\|_1 \leq 1}(P_{(t+1)}X + U_{(t)}), & \text{(10b)} \\[2mm]
U_{(t+1)} \leftarrow U_{(t)} + P_{(t+1)}X - S_{(t+1)}. & \text{(10c)}
\end{cases}
$$

Finally, we notice that formulation (8) and the derived L-ADMM algorithm are invariant under linear transformation of columns of $X$, meaning if we replace $X$ with $\widetilde{X} = GX$ in (8), where $G$ is a $k \times k$ invertible matrix, then every iterate $P_{(t)}$ is uniquely mapped to $P_{(t)}G^{-1}$ (while $S_{(t)}$ and $U_{(t)}$ are exactly the same), and the objective value has a constant difference $-\log|\det P_{(t)}G^{-1}| = -\log|\det P_{(t)}| + \log|\det G|$. This observation allows us to preprocess the data matrix $X$ by orthogonalizing its rows, so that the update for $P$ in (10a) can be further simplified. The overall algorithm is summarized in Algorithm 1. We empirically found that setting $\rho = nk$ works very well in practice.

---

**Algorithm 1** Solving (7) with L-ADMM

1: take the QR factorization of $X^\top = QR$
2: set $\rho = nk$ and initialize $P_{(0)}$
3: **for** $t = 0, 1, 2, \ldots$ until convergence **do**
4:      $P_{(t+1)} \leftarrow (S_{(t)} - U_{(t)})Q + (1/\rho)P_{(t)}^{-\top}$
5:      $S_{(t+1)} \leftarrow \text{Proj}_{\|\cdot\|_1 \leq 1}(P_{(t+1)}Q^\top + U_{(t)})$
6:      $U_{(t+1)} \leftarrow U_{(t)} + P_{(t+1)}Q^\top - S_{(t+1)}$
7: **end for**
8: **return** $P_{(t)}R^{-\top}$

---

# 5  Experiments

We now provide some numerical experiments to showcase the effectiveness of the proposed volume optimization formulation using the two algorithms described in §4. All the experiments are conducted in MATLAB.

## 5.1  Optimization performance

We synthetically generate random problems, and show the surprisingly effective performance of the two algorithms proposed in §4. For $k = 20$ and $n = 1000$, we randomly generate the groundtruth sparse coefficient matrix $S^\natural$ according to the Bernoulli-Gaussian model with $p = 0.5$, and the groundtruth dictionary matrix $A^\natural$ completely random, and generate the data matrix $X = A^\natural S^\natural$. Due to Theorem 3.4, we know it is identifiable with very high probability, despite $n$ being not that big compared to the number of atoms $k$. Matrix $X$ is used as input to the L-ADMM algorithm as described in Alg. 1. Although Problem (7) is nonconvex, as long as it is identifiable, we know the global optimum is attained at $P^\star = D(A^\natural)^{-1}$, where $D$ is a diagonal matrix with the $i$th diagonal equal to the inverse of the $\ell_1$ norm of the $i$th row of $S^\natural$. As a result, $-\log|\det D(A^\natural)^{-1}|$ is the optimal value for Problem (7) as long as the model is identifiable, and we shall see whether the proposed algorithm is able to attain that optimal value. On the other hand, since L-ADMM directly tackles formulation 8, it is not guaranteed that $P_{(t)}$ is feasible in every iteration, which makes little sense to check the difference $-\log|\det P_{(t)}| + \log|\det P^\star|$. We instead check the optimality gap of the Lagrangian function values, using the optimal dual variable $\Lambda$, since we have

$$-\log|\det P| + \text{Tr}(PX - S)\Lambda^\star \geq -\log|\det P^\star|,$$

for any $P$ and a feasible $S$. Obviously, the gap equals to zero when $P = P^\star$ and $S = S^\star$, in which case $P^\star X - S^\star = 0$. Furthermore, it is easy to show that an optimal $\Lambda$ is $(\widetilde{S}^\natural)^\dagger$. In this simulation with known groundtruth factors, we will use this to measure the optimality gap.

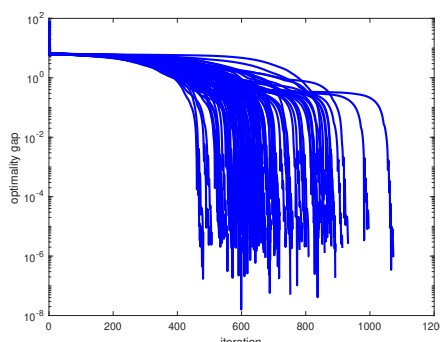

The convergence behavior of 100 random trials of the L-ADMM Algorithm 1 are shown in Figure 2. The somewhat surprising observation is that it achieves essentially zero optimality gap in all instances, which is somewhat surprising as we are, after all, trying to solve NP-hard problems. Among the 100 random trials, L-ADMM usually requires approximately 500–1000 iterations to converge; together with the low-complexity iterations described in Algorithm 1, it usually takes just a few seconds to execute a trial. For a dictionary learning problem of this size, our proposed algorithm takes a lot less time than most existing state-of-the-arts. Some more detailed comparisons can be found in the supplementary material.

Figure 2: 100 random trials of the proposed L-ADMM for dictionary learning as in Algorithm 1. Since we know the groundtruth dictionary is the (essentially) unique minimizer, we can use it to calculate the optimality gap. We see that the optimality gap goes to zero in all random instances.

## 5.2  Dictionary recovery

Next we fix the sample size $n = 1000$ and vary the dictionary size as well as the sparsity parameter $p$ in the Bernoulli-Gaussian model and check how often do we get exact recovery of the groundtruth dictionary $A^\natural$ up to column permutation and scaling. In the previous experiment we simply check the optimality gap between the algorithm output and $-\log|\det D(A^\natural)^{-1}|$, which is also a pretty good indicator of exact recovery since the only ambiguity is an orthogonal rotation, which is very unlikely to exist. Nevertheless, to ensure definitively that the dictionary is recovered as $P^{-1}$, we will normalize the column and use the Hungarian algorithm [Kuhn, 1955] to find the best column matching, and then calculate the estimation error. We declare success if the estimation error is smaller than 1e-5. The results are shown in Figure 3, which agrees with the bound in Theorem 3.4.

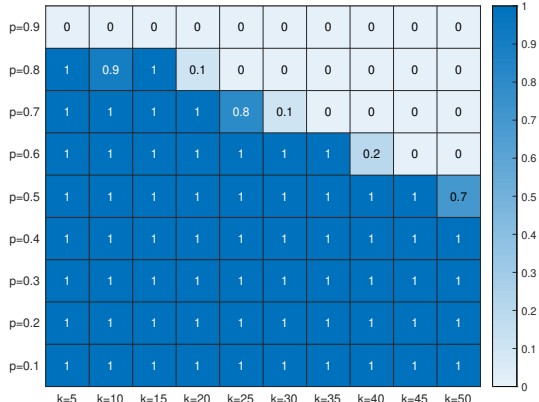

Figure 3: The probability of exactly recovering the dictionary for various dictionary size $k$ and sparsity $p$ (probability of nonzeros in $S^{\natural}$). The sample size is fixed with $n = 1000$.

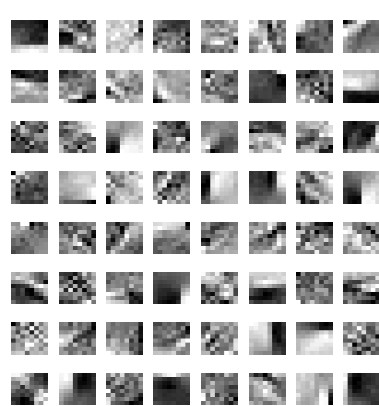

Figure 4: Learned dictionary from a natural image.

We would like to emphasize that compared to similar figures that have appeared in the literature, the groundtruth dictionaries that we try to identify are far from being orthogonal, nor do we care to check whether they satisfy the incoherence assumption; moreover, the recovery is of the *entire* dictionary, not one of the atoms. More comparisons with some existing algorithms can be found in the supplementary.

### 5.3 Real data experiment

We conclude this section with a classical application of dictionary learning on image compression. For a given image, it is first divided into $8 \times 8$ non-overlapping patches, reshaped into a vector in $\mathbb{R}^k$ with $k = 64$, and stacked as columns of the data matrix $X$. After feeding it into the L-ADMM Algorithm 1, the output is inverted to obtain an estimate of the dictionary $A$. Each column of $A$ is then reshaped back into a $8 \times 8$ patch to be shown as an atom for the dictionary. For a monochrome image with $512 \times 512$ pixels, it would be reshaped into a data matrix $X$ of size $64 \times 4096$. Thanks to the high-efficiency of the proposed L-ADMM algorithm, learning a dictionary on a data set of this size takes only a few minutes from a random initialization. The resulting dictionary learned from a natural image is shown in Figure 4. The result agrees with most other dictionary learning algorithms, with some atoms representing edges and bright spots, while some others resemble a few DCT bases.

## 6 Conclusion

In this paper we aimed at answering the question of when dictionary learning is identifiable, with a specific requirement that the formulation uses the $\ell_1$ norm to promote sparsity. With a novel formulation that minimizes the volume of the dictionary subject to $\ell_1$ constraints, we close the case with an affirmative yes, complete dictionary learning is indeed *globally identifiable* under mild conditions. Compared to similar works, our result is global, requires no conditions such as incoherence on the complete dictionary, and at the same time achieves the same sample complexity bound when the sparse coefficients satisfy the Bernoulli-Gaussian model. Finally, we propose algorithms to solve the volume optimization problem, specifically one based on linearized ADMM in the main paper, and demonstrate their exceptional performances on both synthetic and real data.

### Acknowledgments and Disclosure of Funding

This work is supported in part by NSF ECCS-2237640.

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

# A Proof of Theorem 3.2

The proof uses the notion of polar sets from convex analysis [Rockafellar, 1970]:

**Definition A.1** (Polar set). Given a set $C$ that contains the origin, its polar set, denoted as $C^\circ$, is defined as

$$C^\circ = \{x \mid x^\top y \leq 1, \forall y \in C\}.$$

As an example, the polar of the unit ball $\mathcal{B} = \{x \mid \|x\| \leq 1\}$ is itself, i.e., $\mathcal{B}^\circ = \mathcal{B}$, using the Cauchy-Schwartz inequality. Due to the definition of dual norm, we also have immediately that the polar set of any norm ball is the norm ball defined by its dual norm, e.g., the $\ell_\infty$ norm ball $[-1, 1]^n$ is the polar of the $\ell_1$ norm ball $\{x \mid \|x\|_1 \leq 1\}$. Given a polyhedron in the form of conv($H$), i.e., the convex hull of a set of vectors given as the columns of $H$, then its polar is a polyhedron in the form of linear inequalities:

$$\text{conv}(H)^\circ = \{x \mid H^\top x \leq 1\}.$$

The important property of the polar sets is that if two sets satisfy $\mathcal{A} \subseteq \mathcal{B}$, then $\mathcal{B}^\circ \subseteq \mathcal{A}^\circ$, which will be crucial in the sequel. The proof is not hard and can be found in standard textbooks such as [Rockafellar, 1970].

Before proving Theorem 3.2, we show the following lemma as intermediate step:

**Lemma A.2.** *If* conv($\widetilde{S}^\natural V$) *is sufficiently scattered in the hypercube as described in Assumption 3.1, then the solution set of problem* (5) *is* $\{\pm e_1, \ldots, \pm e_k\}$.

*Proof.* As explained in the main paper, the single constraint $\|w^\top \widetilde{S}^\natural\|_1 \leq 1$ is equivalent to a set of $2^n$ linear inequalities

$$w^\top \widetilde{S}^\natural V \leq I^\top,$$

where $V \in \{\pm 1\}^{n \times 2^n}$ contains all $2^n$ vectors $\{\pm 1\}^n$ as its columns. This means $w \in \text{conv}(\widetilde{S}^\natural V)^\circ$, i.e., $w$ belongs to the polar set of conv($\widetilde{S}^\natural V$). The sufficiently scattered condition first assumes that $\mathcal{B} \subseteq \text{conv}(\widetilde{S}^\natural V)$, so conv($\widetilde{S}^\natural V)^\circ \subseteq \mathcal{B}$ since $\mathcal{B}^\circ = \mathcal{B}$. We immediately have that any feasible $w$ for (5) must satisfy $\|w\| \leq 1$. Obviously, letting $w = \pm e_j$ is feasible and attains the maximum.

The second assumption is that the intersection of $\partial \mathcal{B}$ and $\partial \text{conv}(\widetilde{S}^\natural V)$, the boundary of the two sets, is the finite set $\{\pm e_1, \ldots, \pm e_k\}$. This, together with the first assumption, implies that for any direction $y$ not along the coordinates, there exists a point along that direction with norm strictly bigger than 1 that still lie in conv($\widetilde{S}^\natural V$), i.e., for any $y \neq \pm e_j$ and $\|y\| = 1 + \epsilon$ with sufficiently small $\epsilon > 0$, there exists $\theta \in \Delta^n$ such that $y \in \text{conv}(\widetilde{S}^\natural V)$. Then $w$ must also satisfy $w^\top y \leq 1$. Now consider $w = y/\|y\|$, we would have that $\|w\| = 1$ but it's infeasible since $w^\top y = \|y\| > 1$. This excludes any vector with norm 1 that is not in $\{\pm e_1, \ldots, \pm e_k\}$ to be optimal for (5). $\quad\square$

*Proof of Theorem 3.2.* Consider the rescaled groundtruth factors $(\widetilde{A}^\natural, \widetilde{S}^\natural)$ and an optimal solution to (1) $(A^\star, S^\star)$. Obviously, both of them are feasible:

$$X = \widetilde{A}^\natural \widetilde{S}^\natural = A^\star S^\star, \qquad \|\widetilde{S}^\natural_{j,:}\|_1 \leq 1, \|S^\star_{j,:}\|_1 \leq 1, \quad j = 1, \ldots, k.$$

Since $A^\star$ is invertible, we have that $S^\star = (A^\star)^{-1}X = (A^\star)^{-1}\widetilde{A}^\natural \widetilde{S}^\natural$. Now define $W = (A^\star)^{-1}\widetilde{A}^\natural$. Since $\widetilde{A}^\star$ is optimal for (1), we have that

$$|\det A^\star| \leq |\det \widetilde{A}^\natural| \quad \Rightarrow \quad |\det(A^\star)^{-1}\widetilde{A}^\natural| \geq 1 \quad \Rightarrow \quad |\det W| \geq 1. \tag{11}$$

On the other hand, $S^\star = W\widetilde{S}^\natural$ being feasible means

$$\|w_j^\top \widetilde{S}^\natural\|_1 \leq 1, \quad j = 1, \ldots, k,$$

i.e., each row of $W$ satisfies the constraint set of (5). Since $\widetilde{S}^\natural V$ satisfies the sufficiently scattered condition, Lemma A.2 implies that

$$\|w_j\| \leq 1, \quad j = 1, \ldots, k.$$

As a result, using the Hadamard inequality we have

$$|\det \boldsymbol{W}| \le \prod_{j=1}^{k} \|\boldsymbol{w}_j\| \le 1, \tag{12}$$

and the supremum is attained when each row of $\boldsymbol{W}$ is $\pm\boldsymbol{e}_j$. Combining (11) and (12) shows that $|\det \boldsymbol{W}| = 1$ and each row of $\boldsymbol{W}$ is $\pm\boldsymbol{e}_j$. Furthermore, $|\det \boldsymbol{W}|$ is nonzero only if none of the rows is collinear to each other, therefore $\boldsymbol{W}$ must be a permutation matrix times a diagonal matrix with $\pm 1$ on its diagonals. This guarantees identifiability of $\widetilde{\boldsymbol{S}}^{\natural}$ (and $\widetilde{\boldsymbol{A}}^{\natural}$) up to permutation and sign ambiguity, which is equivalent to identifiability of $\boldsymbol{A}^{\natural}$ and $\boldsymbol{S}^{\natural}$ up to permutation and scaling. $\qquad\square$

## B  Proof of Theorem 3.4

We assume that $\boldsymbol{S} \sim \mathcal{BG}(p)$, and the first thing we do is rescaling its rows to have unit $\ell_1$ norms and use it for the problem 5. To simplify the analysis, we can instead directly maximize $\|\boldsymbol{w}\|^2$ subject to $\|\boldsymbol{w}^\top\boldsymbol{S}\|_1 \le 1$, and compare it with the largest $\ell_1$ norm of the rows of $\boldsymbol{S}$. The complement of the intended probability can be bounded as

$$\Pr\left[\sup_{\|\boldsymbol{w}^\top\tilde{\boldsymbol{S}}\|_1 \le 1} \|\boldsymbol{w}\| \le 1\right] \ge \Pr\left[\sup_{\|\boldsymbol{w}^\top\boldsymbol{S}\|_1 \le 1} \|\boldsymbol{w}\| \le \alpha \cap \max_j \|\boldsymbol{S}_{j,:}\|_1 \ge \alpha\right],$$

with arbitrary choice of $\alpha$. Conversely,

$$\Pr\left[\sup_{\|\boldsymbol{w}^\top\tilde{\boldsymbol{S}}\|_1 \le 1} \|\boldsymbol{w}\| > 1\right] \le \Pr\left[\sup_{\|\boldsymbol{w}^\top\boldsymbol{S}\|_1 \le 1} \|\boldsymbol{w}\| > \alpha \cup \max_j \|\boldsymbol{S}_{j,:}\|_1 < \alpha\right]$$

$$\le \Pr\left[\sup_{\|\boldsymbol{w}^\top\boldsymbol{S}\|_1 \le 1} \|\boldsymbol{w}\| > \alpha\right] + \Pr\left[\max_j \|\boldsymbol{S}_{j,:}\|_1 < \alpha\right] \tag{13}$$

where the second inequality is obtained from the union bound. The rest of this section is dedicated to bounding the above two terms. Both of these results rely on the following version of the Bernstein inequality [Bennett, 1962]:

**Theorem B.1** (Bernstein's inequality). *Let $Z_1, \ldots, Z_n$ be independent random variables with $\mathrm{E}[Z_i^2] \le v^2$ and there exists some constant $c$ such that for all integer $d > 2$*

$$\mathrm{E}[|Z_i|^d] \le \frac{1}{2}d!v^2c^{d-2}. \tag{14}$$

*Then*

$$\Pr\left[\left|\sum_{i=1}^{n}(Z_i - \mathrm{E}[Z_i])\right| > \epsilon\right] \le 2\exp\left(-\frac{\epsilon^2}{2(nv^2 + c\epsilon)}\right)$$

**Lemma B.2** (Bounding the second term in (13)). *Suppose $\boldsymbol{S} \in \mathbb{R}^{k \times n}$ is generated from the Bernoulli-Gaussian model $\mathcal{BG}(p)$. Then*

$$\Pr\left[\max_j \|\boldsymbol{S}_{j,:}\|_1 < np(\sqrt{2/\pi} - \epsilon)\right] \le 2k\exp\left(-\frac{np\epsilon^2}{2 + \sqrt{2}\epsilon}\right)$$

*Proof.* Let $\boldsymbol{s} = (s_1, \ldots, s_n)$ be generated from $\mathcal{BG}(p)$, i.e., each $s_i = b_i g_i$ with $b_i$ Bernoulli with probability $p$ and $g_i \sim \mathcal{N}(0, 1)$, we will use Bernstein's inequality with $Z_i = b_i|g_i|$. The moments of $b_i$ all equal to $p$. The random variable $|g_i|$ follows a Chi-distribution of degree 1 so its moments are

$$\mathrm{E}[|g_i|^d] = 2^{d/2}\frac{\Gamma\left(\frac{d+1}{2}\right)}{\Gamma\left(\frac{1}{2}\right)}. \tag{15}$$

For $d = 1$ and 2 we have $\mathrm{E}[Z_i] = p\sqrt{2/\pi}$ and $\mathrm{E}[Z_i^2] = p$. Using the recurrence relation for the Gamma function $\Gamma(t+1) = t\Gamma(t)$ and $\sqrt{2}/\Gamma(1/2) = \sqrt{2/\pi} < 1$ we can bound the rest of the moments with $d > 2$ as

$$\mathrm{E}[|Z_i|^d] \le p\frac{d!}{2^{d/2}}. \tag{16}$$

Therefore the moments satisfy (14) with $c = 1/\sqrt{2}$, results in

$$\Pr\left[\|\boldsymbol{s}\|_1 < np(\sqrt{2/\pi} - \epsilon)\right] \le \Pr\left[\left|\sum_{i=1}^{n}(Z_i - \mathrm{E}[Z_i])\right| > np\epsilon\right]$$

$$\le 2\exp\left(-\frac{n^2p^2\epsilon^2}{2(np + np\epsilon/\sqrt{2})}\right)$$

$$= 2\exp\left(-\frac{np\epsilon^2}{2 + \sqrt{2}\epsilon}\right).$$

Finally, using the union bound

$$\Pr\left[\max_j \|\boldsymbol{S}_{j,:}\|_1 < np(\sqrt{2/\pi} - \epsilon)\right] \le k\Pr\left[\|\boldsymbol{S}_{j,:}\|_1 < np(\sqrt{2/\pi} - \epsilon)\right]$$

$$\le 2k\exp\left(-\frac{np\epsilon^2}{2 + \sqrt{2}\epsilon}\right).$$

$\square$

We now proceed to bound the first term in (13). First we note the following equivalence:

$$\Pr\left[\sup_{\|\boldsymbol{w}^\top\boldsymbol{S}\|_1 \le 1} \|\boldsymbol{w}\| > \alpha\right] = \Pr\left[\inf_{\|\boldsymbol{w}\|=1} \|\boldsymbol{S}^\top\boldsymbol{w}\|_1 < 1/\alpha\right] \tag{17}$$

We are also going to use the following notion of $\delta$-cover from convex geometry [Pisier, 1999] that holds for all $\ell_p$ norm balls, but we are only to instantiate the Eucleadian ball:

**Lemma B.3** ($\delta$-cover). *A finite $\delta$-cover of the unit sphere in $\mathbb{R}^k$ is a finite set $C_\delta$ of points with unit $\ell_2$ norm such that any point on the unit sphere is within $\epsilon$ away from an element in $C_\delta$, i.e.*

$$\min_{\boldsymbol{w}_i \in C_\delta} \|\boldsymbol{w} - \boldsymbol{w}_i\| < \delta, \ \ \forall \|\boldsymbol{w}\| = 1.$$

*For $\delta \in (0, 1)$ there always exists an $\delta$-cover $C_\epsilon$ with cardinality $|C_\delta| < (3/\delta)^k$.*

**Lemma B.4.** *Let $C_\delta = \{\boldsymbol{w}_i\}$ be an $\delta$-cover for the sphere in $\mathbb{R}^k$. Assume that we have both the lowerbound*

$$\|\boldsymbol{S}^\top\boldsymbol{w}_i\|_1 \ge \beta, \forall \boldsymbol{w}_i \in C_\delta$$

*and the upperbound*

$$\|\boldsymbol{S}^\top\|_1 = \sup_{\|\boldsymbol{w}\|_1 \le 1} \|\boldsymbol{S}^\top\boldsymbol{w}\|_1 \le \gamma.$$

*Then*

$$\inf_{\|\boldsymbol{w}\| \le 1} \|\boldsymbol{S}^\top\boldsymbol{w}\|_1 \ge \beta - \gamma\delta\sqrt{k}$$

*Proof.* By definition of the $\delta$-cover, for all $\boldsymbol{w}$ with unit norm we can find $\boldsymbol{w}_i \in C_\delta$ with $\|\boldsymbol{w} - \boldsymbol{w}_i\| < \delta$. Therefore

$$\|\boldsymbol{S}^\top\boldsymbol{w}\|_1 \ge \|\boldsymbol{S}^\top\boldsymbol{w}_i\|_1 - \|\boldsymbol{S}^\top(\boldsymbol{w} - \boldsymbol{w}_i)\|_1 \ge \beta - \|\boldsymbol{S}^\top\|_1\|\boldsymbol{w} - \boldsymbol{w}_i\|_1$$

$$\ge \beta - \|\boldsymbol{S}^\top\|_1\|\boldsymbol{w} - \boldsymbol{w}_i\|\sqrt{k} \ge \beta - \gamma\delta\sqrt{k}.$$

$\square$

**Lemma B.5** (Bounding the first term in (13)). *Suppose $\boldsymbol{S} \in \mathbb{R}^{k \times n}$ is generated from the Bernoulli-Gaussian model $\mathcal{BG}(p)$. Then*

$$\Pr\left[\inf_{\|\boldsymbol{w}\|=1} \|\boldsymbol{S}^\top\boldsymbol{w}\|_1 < np(\sqrt{2/\pi} - \epsilon) - \delta\sqrt{k}np(\sqrt{2/\pi} + \epsilon)\right] \le \left(\left(\frac{3}{\delta}\right)^k + 1\right)2\exp\left(-\frac{np\epsilon^2}{2 + \sqrt{2}\epsilon}\right),$$

*where $\delta \in (0, 1)$ represents any choice of $\delta$-cover for the unit sphere.*

*Proof.* Following Lemma B.4, we have

$$\Pr\left[\inf_{\|\boldsymbol{w}\|\leq 1}\|\boldsymbol{S}^\top\boldsymbol{w}\|_1 < \beta - \gamma\delta\sqrt{k}\right] \leq \sum_{\boldsymbol{w}_i\in C_\delta}\Pr\left[\|\boldsymbol{S}^\top\boldsymbol{w}_i\|_1 < \beta\right] + \Pr\left[\|\boldsymbol{S}^\top\|_1 > \gamma\right], \quad (18)$$

where $C_\delta$ is a $\delta$-cover of the unit sphere with cardinality $|C_\delta| < (3/\delta)^k$ according to Lemma B.3.

The bound to the first term in (18) is almost identical to Lemma B.2. Dropping the subscript of $\boldsymbol{w}_i$, we write

$$\|\boldsymbol{S}^\top\boldsymbol{w}\|_1 = \sum_{i=1}^n\left|\sum_{j=1}^k b_{ij}g_{ij}m_j\right| := \sum_{i=1}^n|Z_i|.$$

Without the absolute value, $Z_i$ is normally distributed with zero mean and variance $\sigma^2 = \sum_j m_j^2 b_{ij}$, so

$$\mathrm{E}[|Z_i|^d] = \mathrm{E}[|\sigma G|^d] = \mathrm{E}[|\sigma|^d]\,\mathrm{E}[|G|^d],$$

where $G$ is a standard normal random variable. Since $\sigma^2 = \sum_j m_j^2 b_{ij} \leq \sum_j m_j^2 = 1$, we have for $d \geq 2$

$$\mathrm{E}[|\sigma|^d] \leq \mathrm{E}[|\sigma|^2] = \mathrm{E}\left[\sum_j m_j^2 b_{ij}\right] = p,$$

where for $d = 1$ we have

$$\mathrm{E}[|\sigma|] \geq \mathrm{E}[|\sigma|^2] = \mathrm{E}\left[\sum_j m_j^2 b_{ij}\right] = p.$$

Furthermore, $|G|$ is chi-distributed with degree 1, whose moments are given in (15), so the moments of $|Z_i|$ again follows (16). Again, the moments satisfy (14) with $c = 1/\sqrt{2}$, results in

$$\Pr\left[\|\boldsymbol{S}^\top\boldsymbol{w}\|_1 < np(\sqrt{2/\pi} - \epsilon)\right] \leq \Pr\left[\left|\sum_{i=1}^n(Z_i - \mathrm{E}[Z_i])\right| > np\epsilon\right]$$

$$\leq 2\exp\left(-\frac{n^2p^2\epsilon^2}{2(np + np\epsilon/\sqrt{2})}\right)$$

$$= 2\exp\left(-\frac{np\epsilon^2}{2 + \sqrt{2}\epsilon}\right). \quad (19)$$

To bound $\|\boldsymbol{S}^\top\|_1$, we recall that this is the $\ell_1$ induced norm for matrix $\boldsymbol{S}^\top$, which is shown to be the maximum of the $\ell_1$ norms of the columns of $\boldsymbol{S}^\top$. This means we can use similar arguments used in Lemma B.2 (but applied to the other direction) to have

$$\Pr\left[\|\boldsymbol{S}^\top\|_1 > np(\sqrt{2/\pi} + \epsilon)\right] = \Pr\left[\max_j\|\boldsymbol{S}_{j,:}\|_1 > np(\sqrt{2/\pi} + \epsilon)\right]$$

$$\leq \Pr\left[\|\boldsymbol{S}_{j,:}\|_1 > np(\sqrt{2/\pi} + \epsilon)\right]$$

$$\leq \Pr\left[\left|\sum_{i=1}^n(Z_i - \mathrm{E}[Z_i])\right| > np\epsilon\right] \leq 2\exp\left(-\frac{np\epsilon^2}{2 + \sqrt{2}\epsilon}\right), \quad (20)$$

where we pick an arbitrary $j \in [k]$ in the second line since this event implies that the maximum $\ell_1$ norm of the rows are lowerbounded, and in the third line each $Z_i = b_i|g_i|$ satisfies (16). The proof is complete by combining (18), (19), and (20) with $\beta = np(\sqrt{2/\pi} - \epsilon)$ and $\gamma = np(\sqrt{2/\pi} + \epsilon)$. □

*Proof of Theorem 3.4.* We first instantiate Lemma B.5 with

$$\delta = \frac{n^2p^2(\sqrt{2/\pi} - \epsilon)^2 - 1}{n^2p^2(\sqrt{2/\pi} - \epsilon)(\sqrt{2/\pi} + \epsilon)\sqrt{k}},$$

which obviously satisfies $\delta < 1$ and would also satisfies $\delta > 0$ if

$$\epsilon < \sqrt{\frac{2}{\pi}} - \frac{1}{np}.$$

Then we have

$$\Pr\left[\inf_{\|\boldsymbol{w}\|=1} \|\boldsymbol{S}^\top \boldsymbol{w}\|_1 < 1/np(\sqrt{2/\pi} - \epsilon)\right] \leq \left(\left(\frac{3}{\delta}\right)^k + 1\right) 2\exp\left(-\frac{np\epsilon^2}{2 + \sqrt{2}\epsilon}\right),$$

Combining (13), (17), and Lemma B.2 with $\alpha = np(\sqrt{2/\pi} - \epsilon)$, we obtain

$$\Pr\left[\sup_{\|\boldsymbol{w}^\top \tilde{\boldsymbol{S}}\|_1 \leq 1} \|\boldsymbol{w}\| > 1\right] \leq \Pr\left[\inf_{\|\boldsymbol{w}\|=1} \|\boldsymbol{S}^\top \boldsymbol{w}\|_1 < 1/\alpha\right] + \Pr\left[\max_j \|\boldsymbol{S}_{j,:}\|_1 < \alpha\right]$$

$$\leq 2\left(k + \left(\frac{3}{\delta}\right)^k + 1\right)\exp\left(-\frac{np\epsilon^2}{2 + \sqrt{2}\epsilon}\right). \tag{21}$$

Further, with

$$\epsilon < \frac{(\sqrt{3}k^{1/4} - 1)\sqrt{2/\pi}}{\sqrt{3}k^{1/4} + 1},$$

where the right hand side is obviously positive, we have

$$\delta = \frac{n^2 p^2 (\sqrt{2/\pi} - \epsilon)^2 - 1}{n^2 p^2 (\sqrt{2/\pi} - \epsilon)(\sqrt{2/\pi} + \epsilon)\sqrt{k}} > \frac{n^2 p^2 (\sqrt{2/\pi} - \epsilon)^2}{n^2 p^2 (\sqrt{2/\pi} + \epsilon)^2 \sqrt{k}} > \frac{3}{k}.$$

We can further relax (21) to

$$\Pr\left[\sup_{\|\boldsymbol{w}^\top \tilde{\boldsymbol{S}}\|_1 \leq 1} \|\boldsymbol{w}\| > 1\right] \leq 2\left(k + (k)^k + 1\right)\exp\left(-\frac{np\epsilon^2}{2 + \sqrt{2}\epsilon}\right) \leq 4(k)^k \exp\left(-\frac{np\epsilon^2}{2 + \sqrt{2}\epsilon}\right)$$

$$\leq 4\exp\left(k\log(k) - np(1 - p)\right),$$

where in the last inequality we simply picked a small-enough $\epsilon$ so that

$$\frac{\epsilon^2}{2 + \sqrt{2}\epsilon} < 1 - p.$$

This completes the proof. □

## C  Additional Algorithms

In this section we introduce two algorithms that directly tackles the formulation (7), repeated here:

$$\underset{\boldsymbol{P}}{\text{minimize}} \quad -\log|\det \boldsymbol{P}| \qquad \text{subject to} \quad \|[\boldsymbol{P}\boldsymbol{X}]_{j,:}\|_1 \leq 1, j = 1, \ldots, k.$$

### C.1  Frank-Wolfe algorithm

Our first attempt is to apply the Frank-Wolfe algorithm to approximately solve (7). The Frank-Wolfe algorithm, also known as the conditional gradient method for constrained optimization [Bertsekas, 1999], iteratively minimizes a linear objective, defined by the gradient at the current iterate, under the same constraint set to determine the search direction and obtain the next iterate via some line search approach along the search direction. For the log-determinant objective in (7), we have that the gradient is $-(\boldsymbol{P}^{-1})^\top$. As a result, the Frank-Wolfe algorithm for (7) is given in Algorithm 2.

Regarding the line search step, we propose to use the backtracking line search (Armijo rule) [Bertsekas, 1999] to guarantee sufficient decrease of the objective function. Since the constraint set of (7) is convex, as long as $\boldsymbol{P}_{(t)}$ is feasible, then $\boldsymbol{P}_{(t+1)}$ is also feasible since it is a convex combination of $\boldsymbol{P}_{(t)}$ and $\boldsymbol{P}_d$, which are by definition feasible. Therefore, the only nontrivial part is to find a feasible initialization $\boldsymbol{P}_{(0)}$. This can be done by optimizing an arbitrary linear objective subject to the same

---

**Algorithm 2** Solving (7) with Frank-Wolfe

---

initialize $\boldsymbol{P}_{(0)}$
**for** $t = 0, 1, 2, \ldots$ until convergence **do**
$\quad \boldsymbol{P}_d = \arg\min\limits_{\boldsymbol{P}} - \text{Tr}(\boldsymbol{P}_{(t)}^{-1}\boldsymbol{P})$
$\qquad$ subject to $\|[\boldsymbol{PX}]_{j,:}\|_1 \leq 1, j = 1, \ldots, k$
$\quad \alpha \leftarrow 1$
$\quad$ **while** $-\log|\det(\boldsymbol{P}_{(t)} + \alpha_t(\boldsymbol{P}_d - \boldsymbol{P}_{(t)}))| >$
$\qquad\qquad -\log|\det \boldsymbol{P}_{(t)}| + (\alpha/2)\,\text{Tr}(\boldsymbol{P}_{(t)}^{-1}(\boldsymbol{P}_d - \boldsymbol{P}_{(t)}))$ **do**
$\qquad \alpha \leftarrow \alpha/2$
$\quad$ **end while**
$\quad \boldsymbol{P}_{(t+1)} = \boldsymbol{P}_{(t)} + \alpha(\boldsymbol{P}_d - \boldsymbol{P}_{(t)})$
**end for**

---

constraint as (7) (which means one should not apply line search at this step). If $\boldsymbol{A}$ is square, there is a simple initialization that works really well in our experience, by setting $\boldsymbol{P}_{(0)}$ as a diagonal matrix with
$$\boldsymbol{P}_{(0)}(c, c) = 1/\|\boldsymbol{X}_{j,:}\|_1,$$
i.e., rescaling each row of $\boldsymbol{X}$ to have unit $\ell_1$ norm.

In terms of complexity, each iteration is dominated by the linear programming with $k^2$ variables. Without exploiting any structure, the per-iteration complexity could be as high as $O(k^6)$. However, the linear programming to be solved in Algorithm 2 is blessed with structures to be exploited to greatly reduce the complexity. Denote $\boldsymbol{p}_i$ as the $i$th row of $\boldsymbol{P}$, then the linear programming in each iteration of Algorithm 2 is in fact $k$ independent problems, each involving only one row of $\boldsymbol{P}$; let $\boldsymbol{f}_i$ denote the $i$th column of $\boldsymbol{P}_{(t)}^{-1}$, then we should solve the following problem with $i = 1, \ldots, k$

$$\min_{\boldsymbol{p}_i} - \boldsymbol{f}_i^\top \boldsymbol{p}_i \quad \text{subject to} \ \ \|\boldsymbol{p}_i^\top \boldsymbol{X}\|_1 \leq 1. \tag{22}$$

Each of these problems involves $k$ variables, which can be solved with $O(k^3)$ flops. This important observation brings the per-iteration complexity of Algorithm 2 down to $O(k^4)$.

Note that the Frank-Wolfe algorithm is also able to handle the overdetermined case when the dictionary matrix $\boldsymbol{A}$ is tall, although this case is rarely considered in the context of dictionary learning: as long as $\boldsymbol{A}$ has full column rank, we can make the change-of-variable $\boldsymbol{P} = \boldsymbol{A}^\dagger$, and the objective of (7) can be replaced by $(1/2)\log(\boldsymbol{PP}^\top)$, with its gradient equals to $-(\boldsymbol{P}^\dagger)^\top$. Everything else follows easily.

We implement the algorithm in MATLAB and use the built-in `linprog` function in MATLAB to solve each of the linear programming sub-problems (22). Dropping the subscripts in (22), the $\ell_1$ norm constraint can be transformed as linear ones using the standard trick:

$$\min_{\boldsymbol{p},\boldsymbol{t}} - \boldsymbol{f}^\top \boldsymbol{p} \quad \text{subject to} \ \ -\boldsymbol{t} \leq \boldsymbol{p}_i^\top \boldsymbol{X} \leq \boldsymbol{t}, \boldsymbol{I}^\top \boldsymbol{t} \leq 1.$$

Using the same setting as in §5.1 but with smaller problem dimension $n = 200$ and $k = 10$, the convergence of 50 random trials are shown in Figure 5.

## C.2 Cyclic row update

Inspired by the row-separable structure of the constraint set in (7), we also attempt to solve (7) via a block coordinate descent-type algorithm that cyclically updates each row. However, we note that this algorithm only works when $\boldsymbol{P}$ is square, unlike the Frank-Wolfe algorithm, which can be extended to the rare overdetermined case when $\boldsymbol{A}$ is tall. We will drop the log term in the objective function of (7) when designing the cyclic row update algorithm in this subsection.

Using the Laplace's formula, we know that $\det \boldsymbol{P}$ is a linear function with respect to the $i$th row of $\boldsymbol{P}$ using the co-factor expansion

$$\det \boldsymbol{P} = \sum_{j=1}^k (-1)^{i+j} P_{ij} \det \boldsymbol{P}_{ij},$$

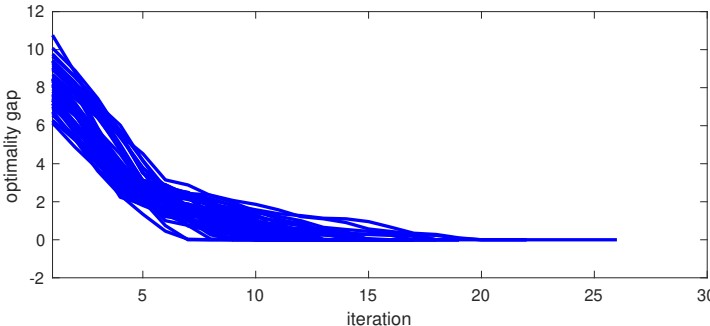

Figure 5: 50 random trials of the Frank-Wolfe Algorithm 2.

where the matrix $\boldsymbol{P}_{ij}$ is obtained by deleting the $i$th row and $j$th column of $\boldsymbol{P}$. Then to apply the block coordinate descent algorithm [Bertsekas, 1999], one would minimize $-|\boldsymbol{f}_i^\top \boldsymbol{p}_i|$ subject to $\|\boldsymbol{p}_i^\top \boldsymbol{X}\|_1 \leq 1$, where $\boldsymbol{f}_i \in \mathbb{R}^k$ is a vector defined according to the co-factor expansion formula.

It looks as if we need to solve two linear programs for each row update, one to maximize $\boldsymbol{f}_i^\top \boldsymbol{p}_i$ and one to minimize $\boldsymbol{f}_i^\top \boldsymbol{p}_i$, but notice that the constraint $\|\boldsymbol{p}_i^\top \boldsymbol{X}\|_1 \leq 1$ is invariant under a sign flip, therefore the only difference between the solutions of the two linear programs would be a sign flip, which is both acceptable due to the inherent sign ambiguity in matrix factorization and dictionary learning in specific. Furthermore, recall that the Cramer's rule shows that

$$\left[\boldsymbol{P}^{-1}\right]_{ji} = (-1)^{i+j} \det \boldsymbol{P}_{ij} \Big/ \det \boldsymbol{P},$$

meaning we can simply define $\boldsymbol{f}_i$ as the $i$th column of $\boldsymbol{P}^{-1}$, and it would not affect the row updates. This makes the definition of $\boldsymbol{f}_i$ exactly the same way as in (22).

---

**Algorithm 3** Solving (7) with cyclic row update

> initialize $\boldsymbol{P}_{(0)}$
> **repeat**
>     **for** $i = 1, \ldots, k$ **do**
>         $\boldsymbol{f} = \boldsymbol{P}^{-1}\boldsymbol{e}_i$
>         $\boldsymbol{p} = \arg\min_{\boldsymbol{p}} -\boldsymbol{f}^\top \boldsymbol{p}$    subject to $\|\boldsymbol{p}^\top \boldsymbol{X}\|_1 \leq 1$
>         replace $i$th row of $\boldsymbol{P}$ with $\boldsymbol{p}$
>     **end for**
> **until** convergence

---

The two proposed algorithms 2 and 3 shows striking similarities, especially considering they both fundamentally solve linear programs in the form of (22) in each iteration. For nonconvex optimization, both algorithms guarantees that every limit point is a stationary point with some additional assumptions, which is reassuring to know. The differences are as follows:

1. Block coordinate descent is guaranteed to monotonically improve the objective function by design, so there is no need for line search as in Frank-Wolfe. This could save some computation as calculating $\det \boldsymbol{P}$ may not be cheap when $k$ is large.
2. On the other hand, each iteration of Frank-Wolfe only need to calculate $\boldsymbol{P}^{-1}$ once for the update of all its $k$ rows, whereas cyclic row update requires to recalculate $\boldsymbol{P}^{-1}$ for each of its row updates. Overall, the per-iteration complexity is almost identical.

An interesting observation is that solving one subproblem (22) resembles the problem formulations in works such as [Spielman et al., 2012, Geng and Wright, 2014, Sun et al., 2016a, Zhai et al., 2020b, Shen et al., 2020] that attempt to find one atom of the dictionary at a time in a deflation manner. The difference is that they are still trying to solve nonconvex problems, and they typically require multiple random initializations due to the risk of finding the same atoms multiple times. From a different perspective, we see that the benefit of using the proposed volume optimization framework provides a

smartly chosen search direction given by rows of $\boldsymbol{P}^{-1}$, which is by definition the current iterate of the dictionary.

This algorithm is again implemented in MATLAB using the built-in `linprog` function. Using the same setting as before with the Frank-Wolfe algorithm, the convergence of 50 random trials are shown in Figure 6. Comparing the two algorithms specifically, we see that Frank-Wolfe succeeds in all 50 trials, while CRU has one failure, although CRU takes much fewer number of iterations to converge.

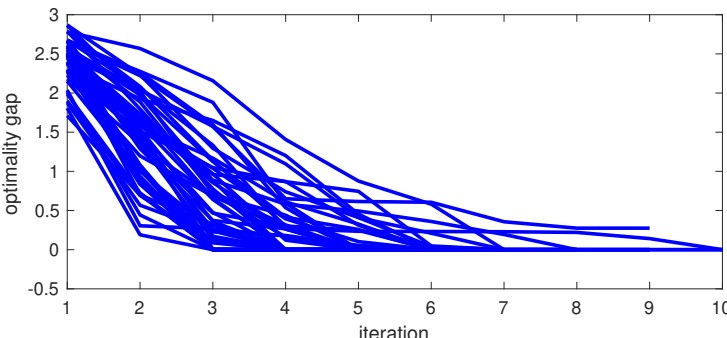

Figure 6: 50 random trials of the cyclic row update Algorithm 3.

## D    Additional Experiments

Here we provide some additional experiments to showcase the performance of the proposed volume-optimization formulation as well as the three proposed algorithms. We start by demonstrating the time-efficiency of the three proposed algorithms to motivate the use of L-ADMM as the go-to method. Then we compare with some state-of-the-art dictionary learning algorithms with identifiability guarantees to showcase the improved performance of dictionary recovery by incorporating the volume criterion.

### D.1    Comparing the three proposed algorithms

In Figures 2, 5, and 6, we showed the per-iteration convergence of L-ADMM, Frank-Wolfe, and cyclic row update (i.e., block coordinate descent [BCD]) as described in Algorithm 1, 2, and 3. While all three algorithms manage to obtain essentially zero optimality gap, L-ADMM takes significantly more number of iterations; on the other hand, it is obvious that the cost of L-ADMM to perform one iteration is much cheaper than those of Frank-Wolfe or BCD. In this subsection we show that, thanks to the simple and efficient per-iteration update of L-ADMM, it is significantly faster than the other two proposed algorithms. We generate synthetic data sets like before, with $n = 1000$ and the sparsity level $p = 0.5$, and set $k = 20$ in Figure 7a and $k = 50$ in Figure 7b. Notice that both the vertical and horizontal axes are in log scale, indicating that L-ADMM is several magnitude faster than the other alternatives. The difference would only become bigger as we increase $n$ and $k$.

### D.2    Dictionary recovery

We compare the dictionary recovery performance of the proposed volume-optimization formulation (using the L-ADMM algorithm in Algorithm 1) with some state-of-the-arts. The most related work are [Gribonval and Schnass, 2010, Wu and Yu, 2017, Wang et al., 2020], which focus on the local identifiability of formulation (2) without proposing new algorithms. For algorithms with claimed identifiability guarantees, the closest we find are the alternating direction method (ADM) by Sun et al. [2016a,b], $\ell_4$-norm maximization ($\ell_4$-max) by Zhai et al. [2020b], and $\ell_3$-norm maximization ($\ell_3$-max) by Shen et al. [2020]. We note that ADM is not the main algorithm that is analyzed in Sun et al. [2016a,b], but it works better and has the benefit of recovering the entire dictionary simultaneously; the analyzed manifold optimization algorithm in [Sun et al., 2016a,b] recovers the

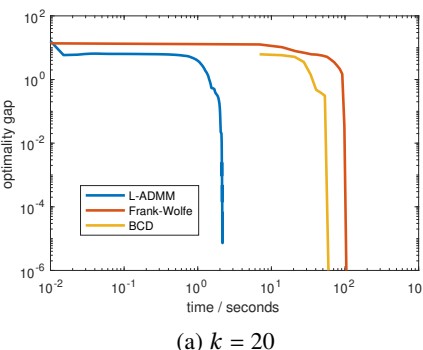
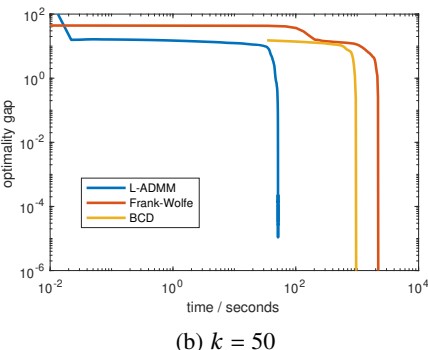

(a) $k = 20$                                    (b) $k = 50$

Figure 7: Convergence (with respect to time in seconds) of the three proposed algorithms. In both cases $n = 1000$ and $p = 0.5$, and we vary the value of $k$. It is clear from both figures that L-ADMM is significantly more efficient than Frank-Wolfe or BCD.

dictionary one atom at a time, which requires multiple initializations to guarantee complete recovery, and such procedures are not provided by the authors on their Github page.

Note that all three baseline algorithms, ADM, $\ell_4$-max, and $\ell_3$-max, require that the groundtruth dictionary to be *orthogonal*. They claim that this is, to some extend, without loss of generality by pre-whitening the data $(XX^\top)^{-1/2}X$. We discover that this claim is only true if the sparse coefficients matrix $S$ follows strictly from the Bernoulli-Gaussian generative model, and that the sample size tends to infinity, in which case $(1/n)SS^\top \to pI$; if this assumption does not hold, such a pre-whitening step is not able to make an arbitrary dictionary orthogonal. As a result, when we use the same experimental setting as in §5.2, none of these algorithms are able to exactly recover the true dictionary in any circumstance.

If we restrict ourselves to orthogonal dictionary recovery, then we obtain some sensible results from the three baseline algorithms, although they still perform significantly worse than our proposed work. What is more, even for exactly orthogonal dictionaries they are not able to recover them with very high accuracy. Here we test them in two cases: In the first case shown in Tables 1 and 2, we fix $n = 1000$ and $k = 20$, and vary the sparsity level $p$ from 0.1 to 0.9; in the other case shown in Tables 3 and 4, we fix $n = 1000$ and $p = 0.5$, and vary the dictionary size $k$ from 5 to 50. The recovered dictionary is again permuted using the Hungarian algorithm, and then we declare a successful recovery if the error is less than a threshold. In each case we present two tables, one with the threshold to be $10^{-5}$ (same as the threshold used in §5.2), and the other with threshold $10^{-2}$. We see that the three baseline algorithms can provide reasonable result with the more loose threshold, especially $\ell_3$-max. Regardless, the proposed volume-optimization formulation with the L-ADMM, performs significantly better than all of them.

Table 1: Probability of success when fixing $n = 1000$ and $k = 20$, threshold $10^{-5}$

| | $p = 0.1$ | $p = 0.2$ | $p = 0.3$ | $p = 0.4$ | $p = 0.5$ | $p = 0.6$ | $p = 0.7$ | $p = 0.8$ | $p = 0.9$ |
|---|---|---|---|---|---|---|---|---|---|
| This work | 1 | 1 | 1 | 1 | 0.9 | 1 | 0.9 | 0 | 0 |
| ADM [Sun et al., 2016a,b] | 1 | 1 | 0.9 | 0.1 | 0 | 0 | 0 | 0 | 0 |
| $\ell_4$-max [Zhai et al., 2020b] | 0 | 0 | 0 | 0 | 0 | 0 | 0 | 0 | 0 |
| $\ell_3$-max [Shen et al., 2020] | 0 | 0 | 0 | 0 | 0 | 0 | 0 | 0 | 0 |

Table 2: Probability of success when fixing $n = 1000$ and $k = 20$, threshold $10^{-2}$

| | $p = 0.1$ | $p = 0.2$ | $p = 0.3$ | $p = 0.4$ | $p = 0.5$ | $p = 0.6$ | $p = 0.7$ | $p = 0.8$ | $p = 0.9$ |
|---|---|---|---|---|---|---|---|---|---|
| This work | 1 | 1 | 1 | 1 | 1 | 1 | 1 | 0.2 | 0 |
| ADM [Sun et al., 2016a,b] | 1 | 1 | 0.9 | 0.2 | 0 | 0 | 0 | 0 | 0 |
| $\ell_4$-max [Zhai et al., 2020b] | 0 | 0 | 0 | 0 | 0 | 0 | 0 | 0 | 0 |
| $\ell_3$-max [Shen et al., 2020] | 1 | 1 | 0.9 | 0.3 | 0 | 0 | 0 | 0 | 0 |

Table 3: Probability of success when fixing $n = 1000$ and $p = 0.5$, threshold $10^{-5}$

| | $k = 5$ | $k = 10$ | $k = 15$ | $k = 20$ | $k = 25$ | $k = 30$ | $k = 35$ | $k = 40$ | $k = 45$ | $k = 50$ |
|---|---|---|---|---|---|---|---|---|---|---|
| This work | 1 | 1 | 1 | 1 | 0.9 | 1 | 1 | 1 | 1 | 0.8 |
| ADM [Sun et al., 2016a,b] | 0.9 | 0.6 | 0.2 | 0 | 0 | 0 | 0 | 0 | 0 | 0 |
| $\ell_4$-max [Zhai et al., 2020b] | 0 | 0 | 0 | 0 | 0 | 0 | 0 | 0 | 0 | 0 |
| $\ell_3$-max [Shen et al., 2020] | 0 | 0 | 0 | 0 | 0 | 0 | 0 | 0 | 0 | 0 |

Table 4: Probability of success when fixing $n = 1000$ and $p = 0.5$, threshold $10^{-2}$

| | $k = 5$ | $k = 10$ | $k = 15$ | $k = 20$ | $k = 25$ | $k = 30$ | $k = 35$ | $k = 40$ | $k = 45$ | $k = 50$ |
|---|---|---|---|---|---|---|---|---|---|---|
| This work | 1 | 1 | 1 | 1 | 1 | 1 | 1 | 1 | 1 | 0.8 |
| ADM [Sun et al., 2016a,b] | 1 | 0.6 | 0.2 | 0 | 0 | 0 | 0 | 0 | 0 | 0 |
| $\ell_4$-max [Zhai et al., 2020b] | 0.7 | 0 | 0 | 0 | 0 | 0 | 0 | 0 | 0 | 0 |
| $\ell_3$-max [Shen et al., 2020] | 1 | 0.9 | 0 | 0 | 0 | 0 | 0 | 0 | 0 | 0 |

