# OpenReview forum: "Global Identifiability of  $\ell_1$-based Dictionary Learning via Matrix Volume Optimization"
_NeurIPS.cc/2023/Conference — NeurIPS 2023 poster_

### Official Review · Reviewer_Fbew · 2023-07-04

**Soundness:** 3 good
**Presentation:** 3 good
**Contribution:** 3 good
**Rating:** 6
**Confidence:** 4

**Summary:**

This article proposes a new method for dictionary learning. The proposed optimisation problem minimises the determinant (volume) is derived from the l1 formulation of dictionary learning. Global identification results are proved under a general assumption of being 'sufficiently scattered'. The case of a Bernoulli-Gaussian model is also studied and recovery with high probability is guaranteed.

On the algorithmic side, the authors describe an algorithm for solving the proposed optimisation problem based on ADMM, and several numerical examples (both synthetic and real data) are presented.

**Strengths:**

The theoretical results are impressive, as the authors establish global identifiability results without the use of the incoherence condition.

**Weaknesses:**

The main concrete example is the Bernoulli-Gaussian setting, which is already covered by previous works, albeit with local guarantees -- It would be good to see more concrete examples on settings that the new framework can handle, but previous results cannot.

There are no comparisons with previous methods in the experiments section of the main paper -- it would be instructive to see comparisons with previous formulations, e.g. (2) to understand if the proposed method does have better recoverability properties, or is simply easier to analyse.

**Questions:**

The equivalence between the different optimisation problems is unclear to me. The main optimisation problem is (1), but the analysis consider (4) and (5). Can you clarify on why these problems are equivalent? You also use phrases such as "conceptually, equivalent" which I find confusing (as I'm not sure whether you actually mean equivalent). To add to this confusion, (4) and (5) seem to assume knowledge of the true factor $S^\sharp$.

Is there a relationship between the 'sufficiently scattered' condition and the incoherence conditions? It would be helpful to see a discussion on why one condition is 'better' than the other.



**Limitations:**

Yes

---

> ### Author Rebuttal · Authors · 2023-08-08
>
> Thank you very much for the positive assessments. Let us address some of your concerns:
>
> > There are no comparisons with previous methods in the experiments section of the main paper
>
> We did compare with some existing methods that claim recovery guarantees in Appendix D.2. The advantage of our proposed method is very significant. Notice that in Tables 1-4, the groundtruth dictionary is in fact orthogonal, which is the most benign case one can have, and we can still give such better performance. When the groundtruth dictionary is not orthogonal, none of the baseline methods can provide any reasonable recovery, unlike our method as shown in $\S5.2$
>
> > Equivalence of formulations (1), (4), and (5).
>
> Let us use an analogy to explain the 'conceptual equivalence' of (1) and (4). Consider the compressive sensing problem given a sensing matrix $A$ and measurements $y=Ax^\natural$ where $x^\natural$ is the unknown sparse vector that we want to recover. In analysis of compressive sensing works, people use the 'conceptual' formulation $\min ||x||_1 \text{ s.t. } Ax=Ax^\natural$ and show that under certain conditions the optimal solution must be $x^\natural$. The idea is similar when we say (1) and (4) are 'conceptually equivalent' in order to show that any optimal solution of (1) must be $A^\natural$ and $S^\natural$ up to permutation and scaling; as it is equivalent to show that any optimal solution of (4) must be a permutation matrix with arbitrary signs.
>
> On the other hand, (4) and (5) are not equivalent. Our claim was that to ensure that any optimal solution of (4) is a signed permutation matrix, a sufficient condition is that any optimal solution of (5) is a signed coordinate vector. This is a sufficient condition, not equivalent.
>
> We hope this clarification is helpful :)
>
> > Is there a relationship between the 'sufficiently scattered' condition and the incoherence conditions?
>
> No. For the dictionary learning model $X=AS$, incoherence is a condition on $A$, while sufficiently scattered is a condition on $S$. Our claim is that DL is globally identifiable if $S$ is sufficiently scattered and $A$ is full rank. Prior work shows that DL is *locally* identifiable if $A$ is incoherent and $S$ follows some probabilistic model such as the Bernoulli-Gaussian model. Overall, our result has a more relaxed condition on $A$ (invertible vs. incoherent), a more relaxed condition on $S$ (sufficiently scattered, which can be implied by the Bernoulli-Gaussian model as we showed in $\S$3.2), and a stronger guarantee (local vs. global).

---

> > ### Comment · Reviewer_Fbew · 2023-08-16
> >
> > Thanks for the clarifications. My rating remains unchanged.

---

### Official Review · Reviewer_Jcds · 2023-07-07

**Soundness:** 3 good
**Presentation:** 3 good
**Contribution:** 3 good
**Rating:** 7
**Confidence:** 4

**Summary:**

The paper studied the classic problem of dictionary learning. A new method is proposed based on optimizing the determinant of the dictionary matrix. Deterministic and probabilistic analysis are conducted for global recovery of the dictionary and coefficients.  Experiments are provided to help demonstrate the effectiveness of the method.

**Strengths:**

The formulation utilizes determinant, which is not common in the current dictionary learning literature.
A global analysis is provided, which does not require the incoherent of the dictionary matrix.

**Weaknesses:**

## Novelty
A crucial related work is Tatli and Erdogan [2021] as the authors pointed out, which studied (1) except that they bound the norm of columns of S while this paper did so for rows of S. What differs in the theoretical analysis? What differs in empirical performance? Please clarify.

## Significance
1. Lack of comparison with other dictionary learning methods
    1. I agree that the proposed method guarantees the correctness of the global solution, which is attractive - but we do not have an algorithm provably converging to the global solution, since the problem is non-convex. On the other hand, Sun et al 2016a/b showed global convergence to a correct solution. Regardless, it would be great if one could compare theoretical results with the literature.
    1. Moreover, it is interesting to see in synthetic experiments how the proposed method compares to baselines.
    1. In particular, since the dictionary is only needed to be invertible rather than incoherent as per Theorem 3.2, why don’t we see some experiments where the dictionary is invertible but coherent?

**Questions:**

## Presentation
1. It appears that linearized ADMM is somewhat standard, so I would suggest maybe some of section 4 could be postponed, and the authors may focus more on the comments above.
1. Multiple optimization problems (1)-(5), (7), (8) have been introduced for different purposes, some of them are equivalent, and some of them are equivalent only under certain conditions. It would be great if the authors could summarize the problems, and clarify the equivalences and purposes.
1. In the remark below Corollary 3.5, it was said the bound works best when p is close to 0.5. Yet, Figure 3 shows that as long as p is <= 0.5. Could you comment on the cause of this apparent discrepancy? Also, you said “this agrees with empirical results in numerous works in dictionary learning”, could you cite which ones?

**Limitations:**

Yes

---

> ### Author Rebuttal · Authors · 2023-08-07
>
> Thank you for the positive assessments. Let us address some of your concerns:
>
> > **Novelty**. A crucial related work is Tatli and Erdogan [2021] as the authors pointed out, which studied (1) except that they bound the norm of columns of S while this paper did so for rows of S. What differs in the theoretical analysis? What differs in empirical performance? Please clarify.
>
> The difference is two-fold:
> 1. In terms of modeling, constraining the norms of the rows of $S$ is without loss of generality, while constraining its columns is not. As we have explained in $\S1.1$, row-scaling of $S$ is inherent in any matrix factorization model. This allows us to consider a generative model that is intrinsically unbounded, such as the Bernoulli-Gaussian model i $\S3.2$, and still guarantee identifiability with high probability. To the best of our knowledge, the analysis by Tatli & Erdogan [2021] is not able to provide such a guarantee for the Bernoulli-Gaussian model, since they impose the bound on the columns of $S$.
> 2. The identifiability conditions are completely different. This was a little surprising to us as well. Tatli & Erdogan's condition is very intuitive: the columns of $S$ are inside the $\ell_1$-norm ball, so they consider the largest Euclidean ball inside the $\ell_1$-norm ball (with radius $1/\sqrt{k}$) and requires the convex hull of $S$ to contain that. In our case, the $\ell_1$-norm constraint is on the rows, so eventually the condition involves a different set of $2^n$ points $SV$, which lie in the $\ell_\infty$-norm ball, and require their convex hull to contain the Euclidean ball with radius $1$. The subsequent analysis turns out to be significantly different.
>
> > Lack of comparison with other dictionary learning methods.
>
> We did compare with some existing methods that claim recovery guarantees in Appendix D.2. The advantage of our proposed method is very significant. Notice that in Tables 1-4, the groundtruth dictionary is in fact **orthogonal**, which is the most benign incoherence one can have, and we can still give such better performance. When the groundtruth dictionary is not orthogonal, none of the baseline methods can provide any reasonable recovery, unlike our method as shown in $\S5.2$.

---

> > ### Comment · Reviewer_Jcds · 2023-08-19
> >
> > Thank you for your effort and response.
> >
> > I was still curious about Significance 1.3
> > > In particular, since the dictionary is only needed to be invertible rather than incoherent as per Theorem 3.2, why don’t we see some experiments where the dictionary is invertible but coherent?
> >
> > I think not requiring the dictionary to be invertible conceptually is a great advance in the literature of DL. It would be great to see some arguments on how much this changes the picture. For example, say one constructs a dictionary as follows. Let us start with an orthogonal matrix $R=[R_1, \dots, R_n]$. The dictionary is constructed by $D=[\theta R_1 + (1-\theta) R_2, R_2, \dots, R_{n}]$. $\theta=1$ is the ideal orthogonal dictionary case, while $\theta=0$ is the degenerate dictionary case. As $\theta \to 0$, the dictionary becomes closer and closer to degenerate, hence more challenging for previous methods in the literature. How would the proposed method behave? What do you trade to have the global solution being the ground-truth one? Do you need more samples, or higher sparsity? Maybe the landscape of the objective becomes worse (note the problem is non-convex), so that it makes an algorithm harder to find global solution?
> >
> > It is fine if the proposed method does not behave "perfectly", I guess just understanding the trade-off here would be of great value to the literature.
> >
> >
> > I am also curious about Question 3
> > > In the remark below Corollary 3.5, it was said the bound works best when p is close to 0.5. Yet, Figure 3 shows that as long as p is <= 0.5. Could you comment on the cause of this apparent discrepancy? Also, you said “this agrees with empirical results in numerous works in dictionary learning”, could you cite which ones?

---

> > > ### Author Response · Authors · 2023-08-21
> > >
> > > Thanks for the insightful idea of designing experiments to test the robustness of the method! We ran a quick test on a case with $n=1000, k=20, p=0.5$ and the recovery is still almost always successful and $\theta$ can be as small as `1e-10`! It did fail when $\theta$ equals to `eps`, which is expected, but even as the original authors we didn't expect the method to be this effective when the conditioning of the dictionary is as bad as $10^{-10}$. The only impact we observe is that it takes longer iterations to converge, but not significantly. We appreciate the insight and would consider putting some experiments along this line in the revised supplementary.
> > >
> > > Regarding the remark below Corollary 3.5, we could probably reword it more precisely. Yes, the bound works the best when $p=0.5$, but that is a bound, and we have not yet established how close the bound is. The fact is, due to inevitable approximations during the derivation of the bound, certain factors may have been omitted, and our guess is that the sample complexity may be best when $p$ is somewhere close to 0. However, we know $p$ cannot be $=0$ (as that would almost surely generate an all-zero $S$, which is clearly not identifiable). We did some experiments with $p=0.01$ and $p=0.001$, and the recovery probability is not longer 100%, which shows that the bound is showing the correct trend. We will revise this comment, and thanks for pointing this out.

---

### Official Review · Reviewer_cSox · 2023-07-09

**Soundness:** 3 good
**Presentation:** 3 good
**Contribution:** 2 fair
**Rating:** 5
**Confidence:** 3

**Summary:**

This paper presents a novel formulation for dictionary learning by minimizing the determinant of the dictionary matrix, subject to the constraint that each row of the sparse coefficient matrix has unit $\ell_1$ norm. The proposed method provides global identifiability guarantees for the ground truth dictionary and sparse coefficient matrices, as long as certain conditions are met. The authors also develop an algorithm based on linearized-ADMM for complete dictionary recovery and provide a probabilistic analysis for the case where the sparse coefficient matrix is generated from the Bernoulli-Gaussian model.

**Strengths:**

1. The paper establishes an identifiability condition on the sparse coefficient matrix and proves that $\ell_1$-based dictionary learning is globally identifiable. This result is a significant contribution, as it extends existing work on identifiability and provides a more general understanding of the problem.

2. The authors introduce an algorithm based on linearized-ADMM for complete dictionary recovery. This algorithm is an important addition to the paper, as it demonstrates a practical approach to solving the new proposed formulation.

**Weaknesses:**

1. A notable limitation of the paper is that the identifiability condition relies on the global minimizers of problem (1), yet the introduced algorithm cannot guarantee obtaining these global minimizers. This concern raises questions about the practical applicability of the theoretical findings, as it may not consistently converge to the desired solution.

2. The paper lacks empirical evidence to support the theoretical results concerning the identifiability condition for dictionary learning. Including experimental observations that corroborate the theoretical findings would strengthen the paper and provide a more convincing argument.

**Questions:**

Please address the two points mentioned in the Weaknesses above.

**Limitations:**

Yes

---

> ### Author Rebuttal · Authors · 2023-08-08
>
> Thank you for the positive assessments. Regarding some of your concerns:
> 1. *Global optimality*. Unfortunately, dictionary learning is known to be a computationally hard problem, and most existing methods that provide guarantee to a global minimizer requires some kind of "good-enough" initialization, which is generally hard to obtain in practice. What we find particularly encouraging is the surprisingly excellent performance of our proposed algorithm in practice, as demonstrated in $\S5.1$ and $\S5.2$, that it almost never fails when the sufficiently scattered condition is satisfied, even with completely random initializations. We will keep on investigating the theoretical analysis on this matter, and hopefully provide some computational guarantees for the proposed method, based on the important stepping stone in this paper.
> 2. *Empirical evidence*. We intended to corroborate the theoretical findings with the experiments in $\S5.2$, which shows the relationship between exact dictionary recovery and the sample size / sparsity level of $S$. In Appendix D.2 we also compared with some existing algorithms that claim performance guarantees, and our proposed method showed significantly better performance in *exactly recovering* the dictionary as well as the sparse coefficient matrix. Notice that the baseline methods in $\S$D.2 fail to recover oblique dictionaries under any circumstances. They were able to recover orthogonal dictionaries to some extent, but not as effective as our method.

---

> > ### Comment · Reviewer_cSox · 2023-08-19
> >
> > Thank you for your response. My concern remains regarding the identifiability condition which relies on the global minimizers of problem (1), especially considering that the introduced algorithm does not guarantee obtaining these global minimizers. Consequently, my rating remains unchanged.

---

### Decision · Program_Chairs · 2023-09-21

**Decision:**

Accept (poster)

**Comment:**

The paper develops a novel formulation for complete dictionary learning based on determinant minimization. This formulation seeks a dictionary matrix A and coefficient matrix S such that the observed matrix X = AS, the rows of S have small L1 norm, and A has small determinant. This determinant regularization helps the authors to prove global uniqueness of the dictionary (up to signed permutation), with no requirement of incoherence, and near optimal rates in random models. [Determinant regularization may also have practical advantages for regularizing the reconstruction problem when the data are nearly degenerate, a common occurrence in practice]. The paper provides theoretical guarantees of identifiability and a practical algorithm based on alternating directions method of multipliers which, while not guaranteed, is shown to be effective in experiments.

Reviewers provided a generally positive evaluation of the paper, expressing appreciation for the paper’s theoretical contributions (identifiability without incoherence, optimal rates). Points of discussion included the lack of a global algorithmic guarantee, effect of incoherence and conditioning, and implications of the experiments. After considering author responses, the reviewers converged to a consensus to accept; the AC concurs and recommends acceptance based on the paper’s novel formulation and theory.